# NMNAT promotes glioma growth through regulating post-translational modifications of P53 to inhibit apoptosis

Jiaqi Liu[1,2], Xianzun Tao[2], Yi Zhu[2], Chong Li[2†], Kai Ruan[2], Zoraida Diaz-Perez[2], Priyamvada Rai[3,4], Hongbo Wang[1]*, R Grace Zhai[2,4]*

[1]School of Pharmacy, Key Laboratory of Molecular Pharmacology and Drug Evaluation (Yantai University), Ministry of Education, Collaborative Innovation Center of Advanced Drug Delivery System and Biotech Drugs in Universities of Shandong, Yantai University, Shandong, China; [2]Department of Molecular and Cellular Pharmacology, University of Miami Miller School of Medicine, Miami, United States; [3]Department of Radiation Oncology, University of Miami Miller School of Medicine, Miami, United States; [4]Sylvester Comprehensive Cancer Center, Miami, United States

**Abstract** Gliomas are highly malignant brain tumors with poor prognosis and short survival. $NAD^+$ has been shown to impact multiple processes that are dysregulated in cancer; however, anti-cancer therapies targeting $NAD^+$ synthesis have had limited success due to insufficient mechanistic understanding. Here, we adapted a *Drosophila* glial neoplasia model and discovered the genetic requirement for $NAD^+$ synthase nicotinamide mononucleotide adenylyltransferase (NMNAT) in glioma progression in vivo and in human glioma cells. Overexpressing enzymatically active NMNAT significantly promotes glial neoplasia growth and reduces animal viability. Mechanistic analysis suggests that NMNAT interferes with DNA damage-p53-caspase-3 apoptosis signaling pathway by enhancing $NAD^+$-dependent posttranslational modifications (PTMs) poly(ADP-ribosyl)ation (PARylation) and deacetylation of p53. Since PARylation and deacetylation reduce p53 pro-apoptotic activity, modulating p53 PTMs could be a key mechanism by which NMNAT promotes glioma growth. Our findings reveal a novel tumorigenic mechanism involving protein complex formation of p53 with $NAD^+$ synthetic enzyme NMNAT and $NAD^+$-dependent PTM enzymes that regulates glioma growth.

*For correspondence:
hongbowangyt@163.com (HW);
gzhai@miami.edu (RGZ)

Present address: †Institute of Molecular Biotechnology of the Austrian Academy of Science (IMBA), Vienna, Austria

Competing interest: The authors declare that no competing interests exist.

## Editor's evaluation

The authors found that NMNAT binds to p53 and that p53 is post-translationally regulated to control apoptosis in glioma models. Work found that depletion of NMNAT1 and NMNAT2 inhibits and that overexpression of the enzymes promotes glioma growth. Furthermore, depletion of $NMNAT_{1/2}$ increases apoptosis and overexpression of the enzymes inhibits apoptosis upon cisplatin treatment. This is an exciting mechanism that extends NAD biology, p53 regulation, and the field of glioma pathogenesis.

## Introduction

Glioma is the most common intrinsic tumor of the central nervous system (CNS) and derives from the neoplastic glial cells or neuroglia (*Goodenberger and Jenkins, 2012*). Based on pathological criteria, gliomas are classified from WHO grade I to IV, among which the high-grade gliomas generally have a much poorer prognosis (*Wesseling and Capper, 2018*). Several major cellular signaling pathways

**eLife digest** One of the most common types of brain cancer, glioma, emerges when harmful mutations take place in the 'glial' cells tasked with supporting neurons. When these genetically damaged cells are not fixed or eliminated, they can go on to multiply uncontrollability. A protein known as p53 can help to repress emerging tumors by stopping mutated cells in their tracks.

Glioma is a highly deadly cancer, and treatments are often ineffective. Some of these approaches have focused on a protein involved in the creation of the coenzyme NAD+, which is essential to the life processes of all cells. However, these drugs have had poor outcomes.

Instead, Liu et al. focused on NMNAT, the enzyme that participates in the final stage of the creation of NAD+. NMNAT is known to protect neurons, but it is unclear how it involved in cancer. Experiments in fruit flies which were then validated in human glioma cells showed that increased NMNAT activity allowed glial cells with harmful mutations to survive and multiply. Detailed molecular analysis showed that NMNAT orchestrates chemical modifications that inactivate p53. It does so by working with other molecular actors to direct NAD+ to add and remove chemical groups that control the activity of p53.

Taken together, these results show how NMNAT can participate in the emergence of brain cancers. They also highlight the need for further research on whether drugs that inhibit this enzyme could help to suppress tumors before they become deadly.

associated with glioma have been well studied, including RTK/Ras/PI3K, p53, and RB signaling pathways (*Cancer Genome Atlas Research Network, 2008*). In addition, metabolism factors, such as IDH1/2, were found to play important roles in glioma (*Yan et al., 2009*). IDH1 is an enzyme of tricarboxylic acid (TCA) cycle in glucose metabolism and the main producer of NADPH (*Molenaar et al., 2014*). However, drugs targeting these pathways showed a limited clinical response, indicating a critical need for the mechanistic understanding of the metabolic requirement for glioma tumorigenesis.

Nicotinamide adenine dinucleotide (NAD$^+$) is an essential signaling cofactor that regulates cancer metabolism through its co-enzymatic function for many bioenergetic pathways, including glycolysis, TCA cycle, and oxidative phosphorylation (*Hanahan and Weinberg, 2011*). Multiple processes associated with NAD$^+$ signaling are dysregulated in cancer, including DNA repair, cell proliferation, differentiation, and apoptosis (*Chiarugi et al., 2012*). Inherited polymorphisms and epigenetic repression of DNA damage repair genes are significantly correlated with the risk of gliomas, indicating that abnormal DNA damage repair plays important roles in glioma formation and progression (*Chen et al., 2010*; *Qi et al., 2017*). One of the key initiation events of DNA damage response is poly (ADP-ribose) polymerase (PARP)-mediated poly(ADP-ribosyl)ation (PARylation), the main process that consumes nuclear NAD$^+$ (*Amé et al., 2004*). Moreover, NAD$^+$-dependent SIRTs-mediated deacetylation regulates many oncogenes and tumor suppressor genes in cancer cells (*Brooks and Gu, 2009*). Consistently, a high level of NAD$^+$ is observed in gliomas (*Reddy et al., 2008*; *Tso et al., 2006*), and 90% of gliomas are susceptible to NAD$^+$ depletion (*Tateishi et al., 2015*). Therefore, it is critical for rapidly proliferating glioma cells to replenish the NAD$^+$ pool for survival.

In the past years, targeting NAD$^+$ metabolism has been considered for cancer therapy, and most efforts have been focused on nicotinamide phosphoribosyltransferase (NAMPT), the rate-limiting enzyme of the NAD$^+$ salvage pathway, whose expression is increased in multiple types of cancer (*Garten et al., 2015*; *Lucena-Cacace et al., 2018*; *Ohanna et al., 2018*; *Pylaeva et al., 2019*). Disappointingly, several clinical trials of NAMPT inhibitors have failed due to low efficacy and high toxicities (*Sampath et al., 2015*), which demands the urgent consideration of an alternative target in the NAD$^+$ metabolic pathway. Nicotinamide mononucleotide adenylyltransferase (NMNAT), the last enzyme in the NAD$^+$ salvage synthetic pathway, has recently emerged as a potential candidate (*Chiarugi et al., 2012*). NMNAT has three isoforms in mammals with distinct subcellular localizations: NMNAT1, in the nucleus; NMNAT2, in the cytosol; and NMNAT3, in the mitochondria (*Berger et al., 2005*). Dysregulations of both NMNAT1 and NMNAT2 have been implicated in cancer. For example, NMNAT1 is considered a poor prognostic marker for renal cancer (*Uhlén et al., 2015*; *Uhlen et al., 2017*). Decreased NMNAT1 expression leads to epigenetic silencing of tumor suppressor genes (*Henderson et al., 2017*). Inhibition of NMNAT1 delays DNA repair and increases rRNA transcription (*Song et al., 2013*). In colorectal cancer, NMNAT2 upregulation correlates with the cancer invasive depth and TNM

stage (*Cui et al., 2016*; *Qi et al., 2018*). In non-small cell lung cancer (NSCLC), NMNAT2 enzymatic activity is upregulated by SIRT3-mediated deacetylation process or p53 signaling (*Li et al., 2013*; *Pan et al., 2014*). Moreover, the depletion of NMNAT2 inhibits cell growth indirectly by reducing glucose availability in neuroblastoma cells (*Ryu et al., 2018*). These observations indicate the regulatory link between compartmentalized NAD$^+$ synthesis and cellular metabolism and rapid cancer cell growth, and further underscore the potential of NMNAT as a viable alternative target in NAD$^+$ synthetic pathway, given their aberrant regulation and critical role in cancer metabolism.

In this report, to address the knowledge gap regarding the role of NMNAT in glioma, we adapted an in vivo glial neoplasia in *Drosophila* (*Read et al., 2009*) and discovered a genetic requirement for NMNAT in glioma growth. Combined with human glioma cell culture models, we characterized the mechanism of NMNAT in gliomagenesis. Our results identified the upregulation of enzymatically active NMNAT as an essential metabolic regulator for promoting gliomagenesis and revealed that NMNAT-sustained PARylation and deacetylation of p53 results in suppression of apoptosis, a key tumor-inhibitory response.

## Results

### NMNAT is upregulated in oncogenic *Ras*$^{v12}$ induced glial neoplasia

The Ras/Raf/ERK signaling cascade is one of the most conserved pathways both in *Drosophila* and human, and a major component of the MAP kinase signaling stress-response network (*Morrison, 2012*). *RAS* mutations are the most commonly found oncogenic alteration in human cancers, most frequently observed in *KRAS* (85%), and to a lesser degree in *NRAS* (12%) and *HRAS* (3%) (*Simanshu et al., 2017*). Upregulated RAS and mutant *RAS* have been detected in gliomas (*Arvanitis et al., 1991*; *Guha et al., 1997*; *Knobbe et al., 2004*; *Rajasekhar et al., 2003*), and activation of Ras has been used to model human glioma in *Drosophila* (*Read, 2011*; *Read et al., 2009*).

*Ras* oncogene at 85D (*Ras85D*) is the *Drosophila* orthologue of human *RAS*. The constitutively active *Ras85D* mutation (G12V), *Ras*$^{v12}$, has been suggested to be analogous to human oncogenic *RAS* mutation and used to induce tumor (*Barbacid, 1987*; *Wu et al., 2010*). We established a *Drosophila* glial neoplasia model by overexpressing *Ras*$^{v12}$ in glial cells, driven by the pan-glial driver repo-GAL4 (*Read et al., 2009*). Green fluorescent protein (GFP) was co-expressed as a reporter to mark the Ras expressing cells. Under normal conditions, the *Drosophila* CNS is wrapped by perineurial, subperineurial, and ensheathing glia (*Freeman, 2015*). Powered with high-resolution quantitative brain morphology analysis (*Brazill et al., 2018b*), we analyzed glial neoplasia tissue using three criteria, (i) tissue double-positive for GFP and endogenous Repo expression; (ii) tissue mass consists of multiple layers of glia of at least 400 cells, and (iii) tissue mass volume greater than 12.4 × 10$^3$ μm$^3$ (*Figure 1—figure supplement 1*). When *Ras*$^{v12}$ was expressed in glia, numerous glial neoplasia tissues marked by GFP and Repo in the brain and ventral nerve cord (VNC) were detected as early as 100 hr after egg laying (AEL), and the volumes of glial neoplasia increased with age (*Figure 1A, B and G*). The brain tumors caused early lethality in pupal stage and greatly reduced survival rate (*Figure 1H*). Notably, compared with the normal brain (*Figure 1C and E*), we found significantly increased endogenous NMNAT in glial cells at both 100 and 150 hr AEL. NMNAT was most prominently increased in the nuclear region (*Figure 1D and F*), suggesting a possible role for NMNAT1, the nuclear isoform, in *Ras*$^{v12}$-induced glial neoplasia formation in *Drosophila*.

### NMNAT is required for glial neoplasia development in *Drosophila*

To determine whether increased NMNAT is required for glial neoplasia development, we used the RNAi approach to downregulate NMNAT expression in *Ras*$^{v12}$-induced glial neoplasia cells (*Brazill et al., 2018a*). NMNAT RNAi-mediated knockdown in Ras$^{v12}$ overexpression cells reduced NMNAT expression level to around 36% of wild-type flies (*Figure 2—figure supplement 1*). Interestingly, knocking down Nmnat drastically reduced both the volume and the number of individual Ras$^{v12}$-expressing glial cells in the brain and VNC at 100 hr AEL (*Figure 2A, C and D*), demonstrating a strong antitumor effect of NMNAT inhibition in vivo. We further analyzed RNAi-mediated knockdown of NMNAT in normal glial cells (without Ras$^{v12}$ expression) and found no growth inhibition (*Figure 2—figure supplement 2*), suggesting NMNAT is not essential for normal cell survival.

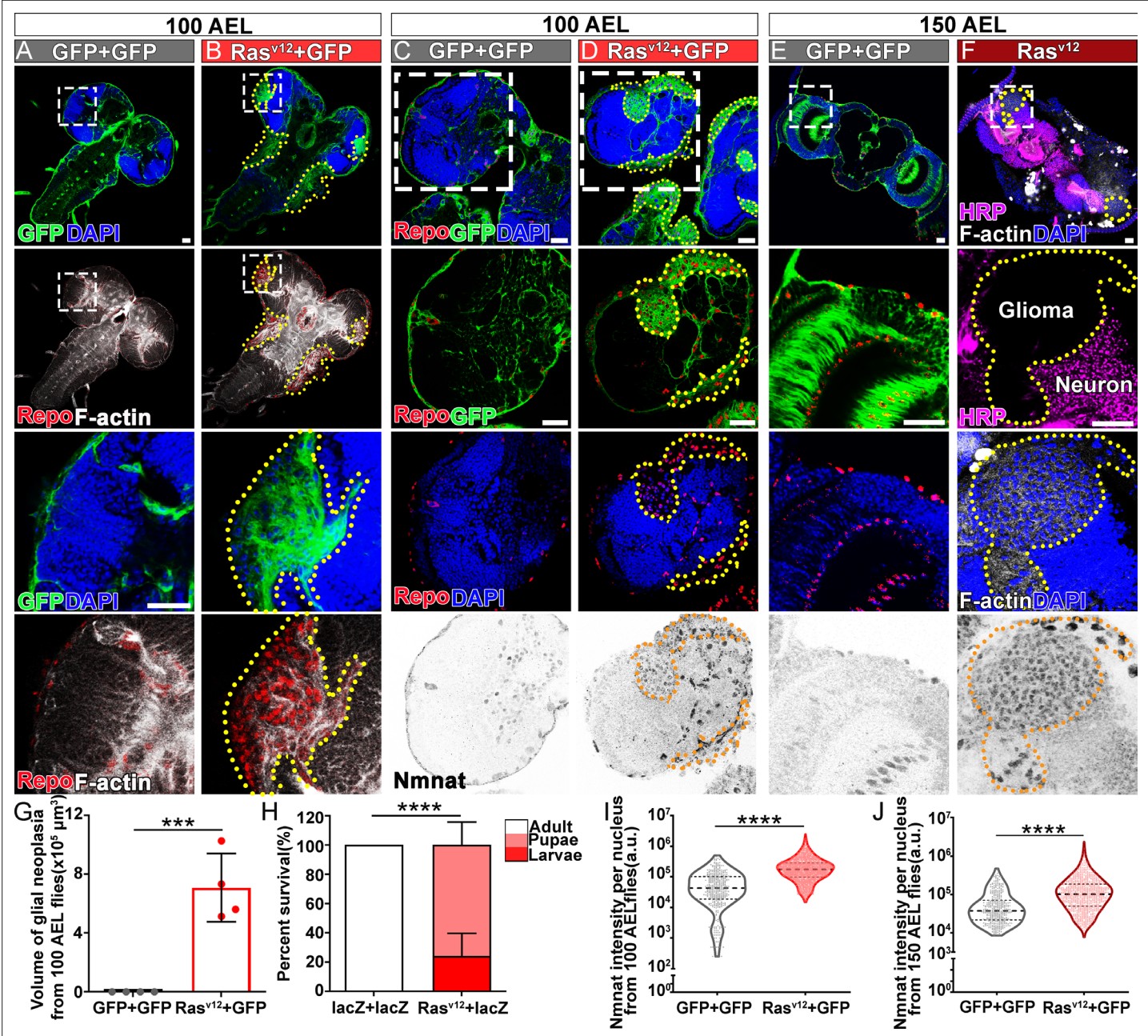

**Figure 1.** NMNAT is upregulated in *Ras^{v12}*-induced glial neoplasia in *Drosophila*. (**A, B**) Larval CNS at 100 AEL with glial expression of GFP+ GFP or Ras^{v12}+ GFP was probed for F-actin (white), Repo (red), and DAPI (blue). The yellow dashed lines mark the boundary of glial neoplasia. The third and fourth rows show the boxed area of the first and second rows. (**C–F**) Larval CNS at 100 (**C, D**) and 150 (**E, F**) AEL. The second to fourth rows show the boxed areas in the first row. (**C–E**) Brains were probed for Nmnat (gray), Repo (red), and DAPI (blue). (**F**) Brains were probed for HRP (magenta), Nmnat (gray), F-actin (white), and DAPI (blue). Yellow dashed lines mark the glial neoplasia boundaries. (**G**) Quantification of the total glial neoplasia volumes in each fly. Data are presented as mean ± s.d., n=4. Significance level was established by one-way ANOVA post hoc Bonferroni test. (**H**) Survival rate. Data are presented as mean ± s.d., n≥3. Significance level was established by Chi-square test. (**I–J**) Nmnat intensity at 100 and 150 AEL. Data are presented as median ± quartiles, n≥3. Significance level was established by one-way ANOVA post hoc Bonferroni test. ***p≤0.001; ****p≤0.0001. Scale bars, 30 μm. AEL, after egg laying; CNS, central nervous system; NMNAT, nicotinamide mononucleotide adenylyltransferase.

The online version of this article includes the following figure supplement(s) for figure 1:

**Figure supplement 1.** Glial neoplasia tissue area in *Drosophila* larval CNS.

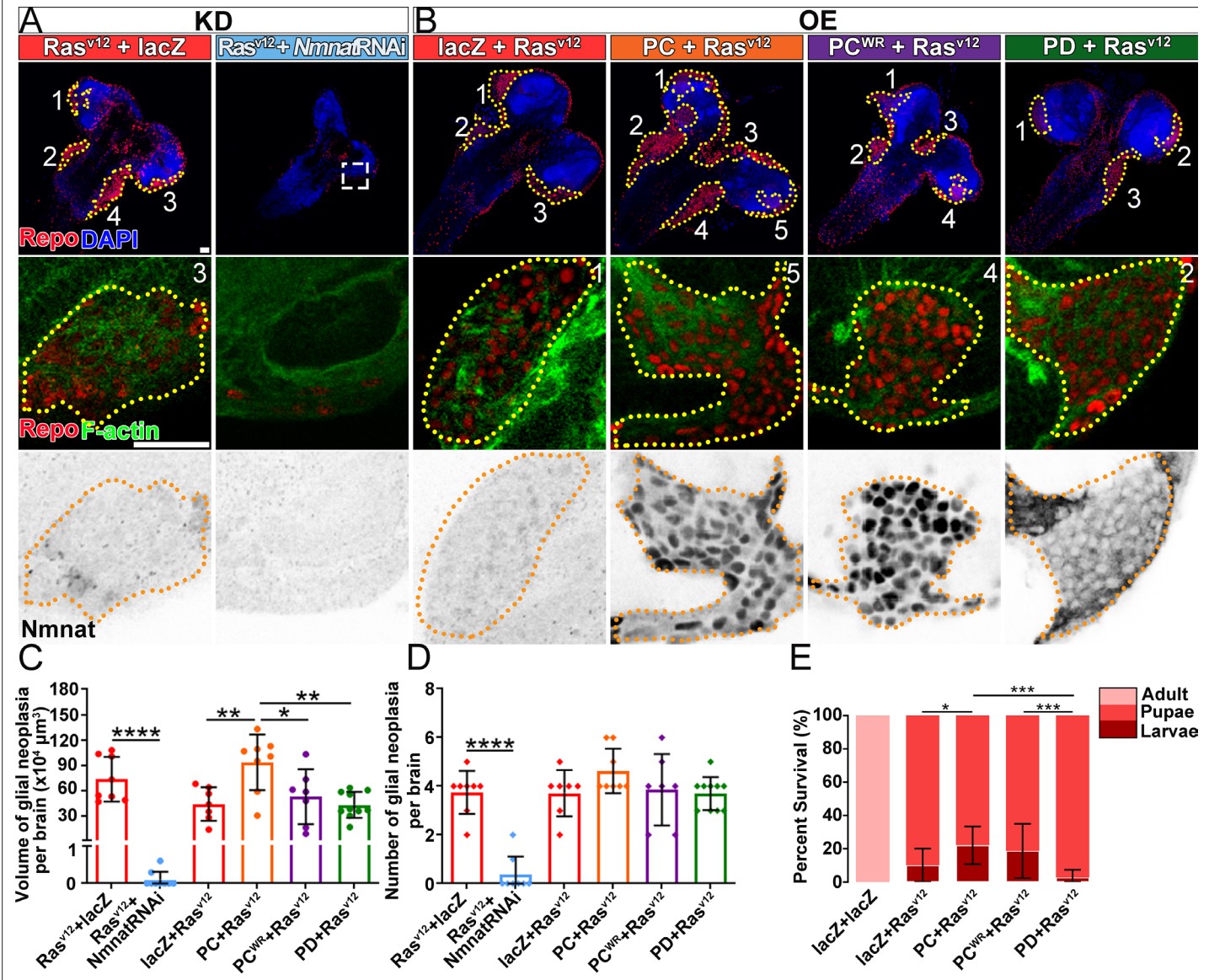

**Figure 2.** NMNAT is required for glial neoplasia growth in *Drosophila*. (**A, B**) Larval CNS at 100 AEL with glial expression of Ras$^{v12}$+lacZ, Ras$^{v12}$+NmnatRNAi, lacZ+Ras$^{v12}$, PC+Ras$^{v12}$, PC$^{WR}$+Ras$^{v12}$, and PD+Ras$^{v12}$ was probed for F-actin (green), Repo (red), DAPI (blue), and Nmnat (gray). Each individual glial neoplasia is marked with dashed lines and numbered. The second and third rows show the high magnification of glial neoplasia areas in the first row. Scale bars, 30 μm. (**C**) Quantification of glial neoplasia volume in each fly. Data are presented as mean ± s.d., n≥7. Significance level was established by one-way ANOVA post hoc Bonferroni test. (**D**) Quantification of glial neoplasia number in each fly. Data are presented as mean ± s.d., n≥7. Significance level was established by one-way ANOVA post hoc Bonferroni test. (**E**) Survival rate of flies with glial expression of Ras$^{v12}$ together with lacZ, PC, PC$^{WR}$, or PD. Data are presented as mean ± s.d., n≥3. Significance level was established by chi-square test. *p≤0.05; **p≤0.01; ***p≤0.001; ****p≤0.0001. AEL, after egg laying; NMNAT, nicotinamide mononucleotide adenylyltransferase.

The online version of this article includes the following figure supplement(s) for figure 2:

**Figure supplement 1.** NMNAT expression is lower in NmnatRNAi fly than wild-type control fly.

**Figure supplement 2.** NMNAT downregulation in glial cells does not affect brain morphology in adult fly.

**Figure supplement 3.** NMNAT-PC$^{WR}$ has no NAD$^+$ synthesis enzyme activity.

Next, we tested whether upregulating NMNAT can promote glial neoplasia formation and growth. *Drosophila* has one *Nmnat* gene, expressing two protein isoforms through alternative splicing, a nuclear isoform Nmnat-PC and a cytosolic isoform Nmnat-PD. The Nmnat-PC (nuclear) and Nmnat-PD (cytoplasmic) isoforms share similar enzymatic activity but are differentially regulated under stress

conditions (*Ruan et al., 2015*). In *Ras^v12*-induced glial neoplasia, dramatically increased Nmnat is mainly observed in the nuclear region (*Figure 1D and F*), likely to be the Nmnat-PC (nuclear) isoform. To further evaluate the compartment-specific role of NMNAT during glial neoplasia formation, we generated flies expressing *Ras^v12* together with Nmnat-PC (nuclear) or Nmnat-PD (cytoplasmic). Consistent with the previous report, Nmnat-PC (nuclear) is highly enriched in the nucleus and colocalizes with the nuclear marker Repo, while Nmnat-PD is predominantly cytoplasmic (*Ruan et al., 2015*). Interestingly, overexpression of Nmnat-PC (nuclear), but not Nmnat-PD (cytoplasmic), significantly increased the total volumes of glial neoplasia (*Figure 2B and C*), while the number of glial neoplasia showed no significant difference among the groups (*Figure 2D*). The lethality of the flies (*Figure 2E*) was positively correlated with glial neoplasia size and overexpression of Nmnat-PC (nuclear) significantly increased the lethality.

To determine whether the enzyme activity of NMNAT is required for glial neoplasia tumorigenesis, we generated flies expressing an enzyme inactive mutant Nmnat-PC (nuclear) isoform (PC^WR) where two key residues for substrate binding were mutated (*Figure 2—figure supplement 3*; *Zhai et al., 2006*). We found that Nmnat-PC^WR (nuclear) overexpression did not significantly affect glial neoplasia volumes or numbers or survival outcome when compared to the control (*Figure 2C–E*). These results suggest that nuclear enzymatically active NMNAT promoted glial neoplasia growth.

## NMNAT is essential to the proliferation of human glioma cells

We next examined the function of NMNAT in human glioma cell proliferation, specifically human NMNAT1 (nuclear) and NMNAT2 (cytoplasmic) (*Berger et al., 2005*). Since approximately 51% of glioma are mutated for p53, we included two glioma cell lines with different p53 status, U87MG with wild-type p53, and T98G with a gain-of-function M237I mutation (*Van Meir et al., 1994*), to dissect common mechanisms of the role of NMNAT in glioma cell growth. We determined NMNAT1 and NMNAT2 protein levels in human glioma cells and normal astroglia cells (SVG p12). Compared to SVG p12 cells, NMNAT1 and NMNAT2 are increased in both glioma cells T98G and U87MG (*Figure 3—figure supplement 3*). Next, we manipulated the expression of NMNAT by siRNA-mediated knockdown and plasmid-mediated overexpression in T98G cells and monitored real-time cell growth using the xCELLigence platform (*Ke et al., 2011*). Interestingly, we found T98G cell proliferation was drastically inhibited when either NMNAT1 or NMNAT2 was knocked down (*Figure 3A*). This observation was confirmed and extended in an MTT assay (*van Meerloo et al., 2011*), where NMNAT1 or NMNAT2 knockdown reduced cell proliferation (*Figure 3—figure supplement 1*). In contrast, overexpressing NMNAT1 or NMNAT2 promoted cell growth (*Figure 3D*). Moreover, we used a plate colony formation assay to determine clonogenic survival (*Franken et al., 2006*), and found that knockdown of NMNAT1 or NMNAT2 reduced the colony numbers of T98G, while overexpression of NMNAT1 or NMNAT2 increased colony formation (*Figure 3B–F*). These results are consistent with the genetic dependency on NMNAT observed in the fly glial neoplasia models, suggesting the conservation of NMNAT function in promoting glioma cell growth and proliferation.

To further determine whether NMNAT is involved in glioma cell survival, we carried out a flow cytometric apoptosis detection assay through flow cytometry. We transfected siRNA targeting NMNAT into T98G cells and then analyzed Annexin V-FITC/PI by flow cytometric 72 hr post-transfection. Interestingly, we found that knockdown of NMNAT, at the knockdown rate of 40–50% for NMNAT1 or at 20–30% for NMNAT2, significantly increased the percentage of apoptotic cells, including early apoptotic and late apoptotic cells (*Figure 3G and H*). We also examined the cell cycle distribution of these cells. The cell cycle assay showed G2/M phase was only slightly increased in T98G cells with NMNAT1 knockdown (*Figure 3—figure supplement 2*). These results suggest that NMNAT promotes glioma cell growth mainly through inhibiting cell apoptosis.

## Overexpression of NMNAT decreases caspase-3 activation in glioma

The cysteine-dependent proteases (caspases) are activated by upstream proteins to mediate apoptosis (*Kurokawa and Kornbluth, 2009*). Caspase-3 is the main effector protease cleaving a large number of substrates during apoptosis. Previous studies revealed that nuclear translocation and accumulation of caspase-3 play a critical role in the progression of apoptosis (*Prokhorova et al., 2018*). The caspase-mediated pathway is highly conserved in mammalian and *Drosophila* (*Fuchs and Steller, 2011*; *Shi, 2001*; *Figure 4—figure supplement 1A*). To validate the role of caspase pathway in *Drosophila* glial

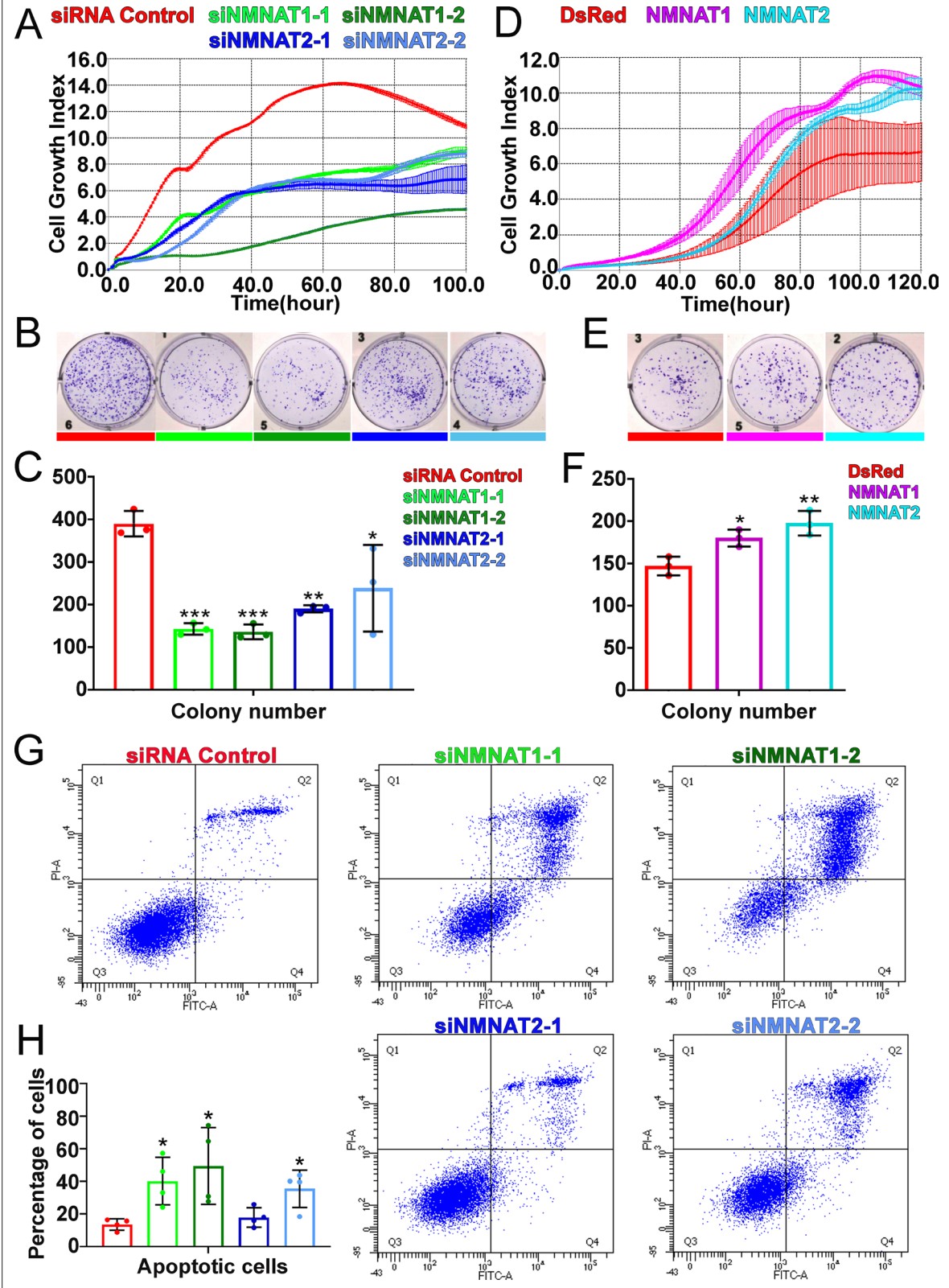

**Figure 3.** NMNAT expression is essential to the proliferation of human GBM cells. (**A, D**) The xCELLigence real-time cell analysis assay was used to monitor the growth index of T98G cells after NMNAT knockdown by transfecting siNMNAT1 or siNMNAT2, or after NMNAT overexpression by transfecting NMNAT1 or NMNAT2 plasmid. Cells transfected with siRNA control or DsRed were used as controls. (**B, E**) Colony formation assay was used to measure the colony formation capabilities of T98G cells after NMNAT knockdown by transfecting siNMNAT1 or siNMNAT2, or after NMNAT

*Figure 3 continued on next page*

*Figure 3 continued*

overexpression by transfecting NMNAT1 or NMNAT2. Cells transfected with siRNA control or DsRed were used as controls. (**C, F**) Quantification of the colony number in (**B, E**). Data are presented as mean ± s.d. n=3. Significance level was established by one-way ANOVA post hoc Bonferroni test. (**G**) T98G cell apoptosis was detected by flow cytometry after NMNAT knockdown. (**H**) Quantification of apoptotic cells rate of siRNA control, siNMNAT1-1, siNMNAT1-2, siNMNAT2-1, and siNMNAT2-2. The sum of Q2 and Q4 was quantified as apoptotic cells. Data are presented as mean ± s.d. n=4. Significance level was established by t-test. *p≤0.05; **p≤0.01; ***p≤ 0.001. GBM, glioblastoma multiforme; NMNAT, nicotinamide mononucleotide adenylyltransferase.

The online version of this article includes the following source data and figure supplement(s) for figure 3:

**Source data 1.** siRNA sequences for NMNAT1/2 knockdown and primer sequences for PCR.

**Figure supplement 1.** T98G cells viability is inhibited after knockdown NMNAT1 or NMNAT2.

**Figure supplement 1—source data 1.** T98G cell viability was inhibited after knockdown NMNAT1 or NMNAT2.

**Figure supplement 2.** Knockdown of NMNAT does not affect cell cycle.

**Figure supplement 3.** NMNAT protein is upregulated in human glioma cells.

**Figure supplement 3—source data 1.** NMNAT protein was upregulated in human glioma cells.

neoplasia, we examined tumor growth in flies with downregulation of DCP1, the homolog of mammalian caspase-3/7. In these flies, glial neoplasia volume was significantly increased (***Figure 4—figure supplement 1B and C***), suggesting the important role of caspase-mediated apoptosis in preventing *Drosophila* glial neoplastic growth. To test whether NMNAT regulates this process, we determined the localization and protein levels of caspase-3 in the glial neoplasms with overexpression of different Nmnat isoforms. We used Repo and DAPI to label the nuclei region and observed a significant decrease of caspase-3 levels in glial neoplasms that overexpress Nmnat-PC (nuclear), compared with those overexpressing lacZ, Nmnat-PC^WR (nuclear), or Nmnat-PD (cytoplasmic) (***Figure 4A and C***). In addition, when we knocked down Nmnat in Ras^v12-expressing glial cells, we observed significant nuclear enrichment of caspase-3 (***Figure 4B and D***). These results suggest that NMNAT is a negative regulator of glial neoplastic cell apoptosis in *Drosophila*.

Next, we examined apoptosis and the activation of caspase-3 in human glioma cells. We found that knockdown of NMNAT led to increased nuclear caspase-3 (***Figure 5A and C***). Western blot analysis showed a specific increase of fully processed P17/19 species of cleaved caspase-3 (***Figure 5D***), indicating the activation of apoptosis (***Porter and Jänicke, 1999***). To examine the effect of overexpressing NMNAT on apoptosis, we employed cisplatin treatment to induce apoptosis as the basal level of apoptosis in T98G glioma cells is low (***Kondo et al., 1995***). Cisplatin significantly increased nuclear caspase-3 levels as expected. Interestingly, overexpression of either NMNAT1 or NMNAT2 reduced nuclear caspase-3 in cisplatin-induced apoptosis (***Figure 5B and E***), specifically the fully processed cleaved caspase species P17/19 as shown by Western blot analysis (***Figure 5F***). Taken together, these results suggest that NMNAT promotes glioma growth by inhibiting caspase-mediated apoptosis.

## Overexpression of NMNAT increases DNA damage tolerance and decreases nuclear p53 in glial neoplasia

DNA instability is one of the hallmarks of cancer. Two common strategies cancer cells use to avoid the triggering cell apoptosis by DNA damage are hyperactivating DNA damage repair, and inactivating cell apoptosis initiation (***Norbury and Zhivotovsky, 2004***). Since NAD^+ plays important regulatory roles in both DNA damage repair and cell apoptosis, and NAD^+ synthase activity is required for glial neoplasia growth (***Figure 2***), we next examined the effect of NMNAT on the DNA damage pathway in glioma. We first determined DNA damage by using a phosphor-specific antibody to histone 2A variant (H2Av), a marker for DNA double-strand breaks (***Lake et al., 2013***). We observed a significant elevation of H2Av signal in Nmnat-PC (nuclear) overexpressing brains compared to that in Nmnat-PD (cytoplasmic), Nmnat-PC^WR (nuclear), or lacZ overexpressing brains (***Figure 6A***), suggesting DNA damage level is higher in glial neoplasia with Nmnat-PC (nuclear) overexpression.

We next examined the distribution of endogenous p53 in glial neoplasia and found that while in control glial neoplasia cells (LacZ group), p53 was relatively evenly distributed with ~40% of p53 in the nucleus, a significantly reduced nuclear p53 pool (~20%) was found in Nmnat-PC (nuclear) overexpressing glial neoplasia cells (***Figure 6B and D***). Together with the observation of higher DNA

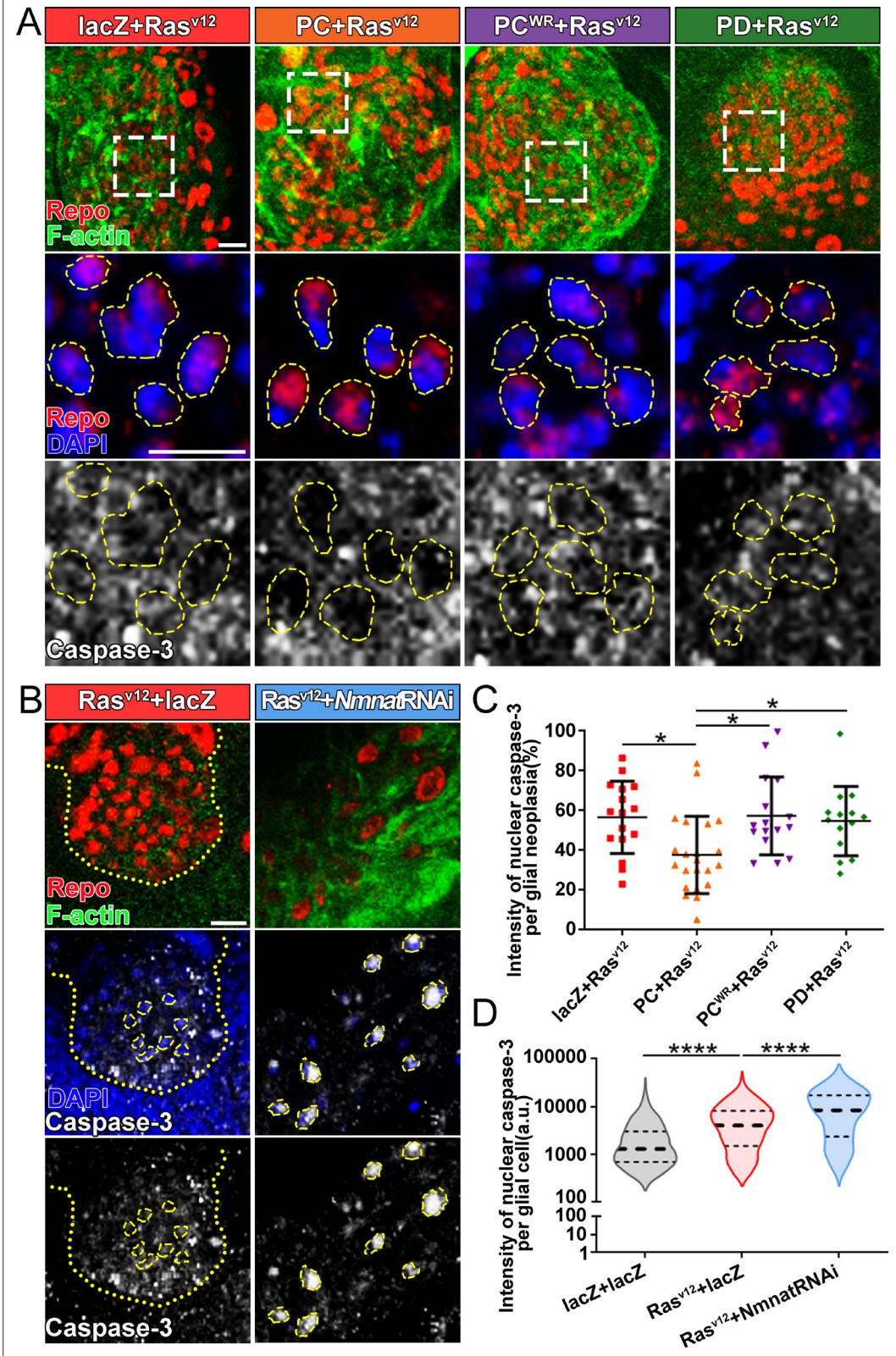

**Figure 4.** Overexpression of NMNAT decreases caspase-3 activation in glial neoplasia. (**A**) Glial neoplasia from files expressing lacZ, PC, PC[WR], or PD were probed for Repo (red), F-actin (green), DAPI (blue), and caspase-3 (gray). The top row shows the whole glial neoplasia area. The second and third rows are the high magnification of the boxed areas in the first row. Yellow dashed lines indicate the nuclear area. (**B**) Glial neoplasia from flies

*Figure 4 continued on next page*

*Figure 4 continued*

expressing lacZ or Nmnat RNAi were probed for Repo (red), F-actin (green), DAPI (blue), and caspase-3 (gray). Yellow dot lines indicate glial neoplasia boundary in the Ras$^{v12}$+lacZ group. Yellow dashed lines indicate the boundaries of the nucleus and cytoplasm. Scale bars, 10 μm. (**C**) Quantification of the percentage of nuclear caspase-3 intensity per glial neoplasia. Data are presented as mean ± s.d. n≥3. Significance level was established by one-way ANOVA post hoc Bonferroni test. (**D**) Quantification of the nuclear caspase-3 per glial cell. Data are presented as median ± quartiles, n≥3. Significance level was established by one-way ANOVA post hoc Bonferroni test. *p≤0.05. ****p≤0.0001. NMNAT, nicotinamide mononucleotide adenylyltransferase.

The online version of this article includes the following figure supplement(s) for figure 4:

**Figure supplement 1.** Blocking caspase pathway in *Ras$^{v12}$* overexpressing fly.

damage levels in Nmnat-PC (nuclear) overexpressing glial neoplasia cells, these results indicate that Nmnat-PC (nuclear) expression potentially regulates p53 response to DNA damage, presumably to allow higher tolerance to DNA damage. p53 is a key player controlling cell fate in response to DNA damage: initiate DNA repair when there is limited DNA damage, and induce apoptosis when DNA damage is too severe (*Roos and Kaina, 2013*). To validate the role of p53 in glial neoplasia development in *Drosophila*, we examined the effect of a p53 inhibitor: pifithrin-α (PFT-α). PFT-α is reported to inhibit translocation of p53 and affect p53-related transactivation (*Komarov et al., 1999*; *Leker et al., 2004*; *Murphy et al., 2004*). We analyzed glial neoplasia tissue volume with GFP and DAPI staining in the CNS of flies (*Figure 7A*). The glial neoplasia volume was significantly increased in PFT-α-treated flies compared to that in DMSO-treated flies (*Figure 7C*). The increase in glial neoplasia volume was accompanied by a decrease in survival (*Figure 7B*), and the reduced cleaved caspase-3 intensity (*Figure 7D*). These results suggest p53 is critical for inhibiting glial neoplastic growth in *Drosophila*, and p53 inhibition phenocopies NMNAT overexpression in glial neoplasia growth. p53 depletion rescues NMNAT knockdown induced caspase-3 activation.

To further assess the role of p53 in NMNAT knockdown-induced apoptosis, we examined the effects of p53 depletion combined with NMNAT knockdown in human glioma cells. We employed two approaches to reduce/deplete p53: siRNA transfection or shRNA lentiviral transduction in both U87MG and T98G cell lines. After p53 depletion, we carried out siNMNAT-mediated knockdown and probed for cleaved caspase-3 to examine the activation of apoptosis in both T98G and U87MG cells. Under all four conditions, two cell types and two modes of p53 depletion, we observed a consistent reduction of siNMNAT-induced apoptosis activation when p53 was depleted. As shown in *Figure 8* (shRNA lentiviral knockdown) and *Figure 8—figure supplement 1* (siRNA knockdown), cleaved caspase-3 expression level in p53 and NMNAT double knockdown cells was reduced compared to those in siNMNAT cells, suggesting that p53 depletion reduced significantly cleaved caspase-3 expression in NMNAT knockdown glioma cells. These results indicate p53 is a key mediator of NMNAT knockdown-induced apoptosis in glioma.

## NMNAT regulates PARylation and acetylation of p53

Our observations that NMNAT overexpression-induced higher tolerance to DNA damage and altered p53 response is intriguing. Maintaining functional DNA repair is critical for cancer cells to survive during rapid cell proliferation and the accompanying constant need for DNA replication. In response to DNA damage, PARP1 catalyzes NAD$^+$-dependent PARylation of a large number of proteins (including p53), a process that is one of the largest NAD$^+$ consumers in the nucleus (*Fischbach et al., 2018*; *Kim et al., 2005*). It has been shown that PARylated p53 has reduced stability and activity (*Simbulan-Rosenthal et al., 1999*). We hypothesize that NMNAT regulates PARylation in glioma cells. To test this hypothesis, we first examined the level of protein PARylation under NMNAT overexpression, and using dot blot analysis, we found that protein PARylation level was significantly increased in NMNAT1 or NMNAT2 overexpressing cells and significantly reduced with siRNA knockdown (*Figure 9A and B* and *Figure 9—figure supplement 1*). Next, we examined the protein-protein interaction among p53, NMNAT1, and PARP1 using immunoprecipitation. Interestingly, we detected PARP1 and NMNAT proteins in the p53-immunoprecipitated fraction (*Figure 9C*). Furthermore, although total p53 levels were not significantly affected by NMNAT expression, the level of PARP1 immunoprecipitated with p53 was increased with NMNAT overexpression (*Figure 9C*). We observed consistent results in U87MG cells that NMNAT interacts with p53 (*Figure 9—figure supplement 2*). These results suggest

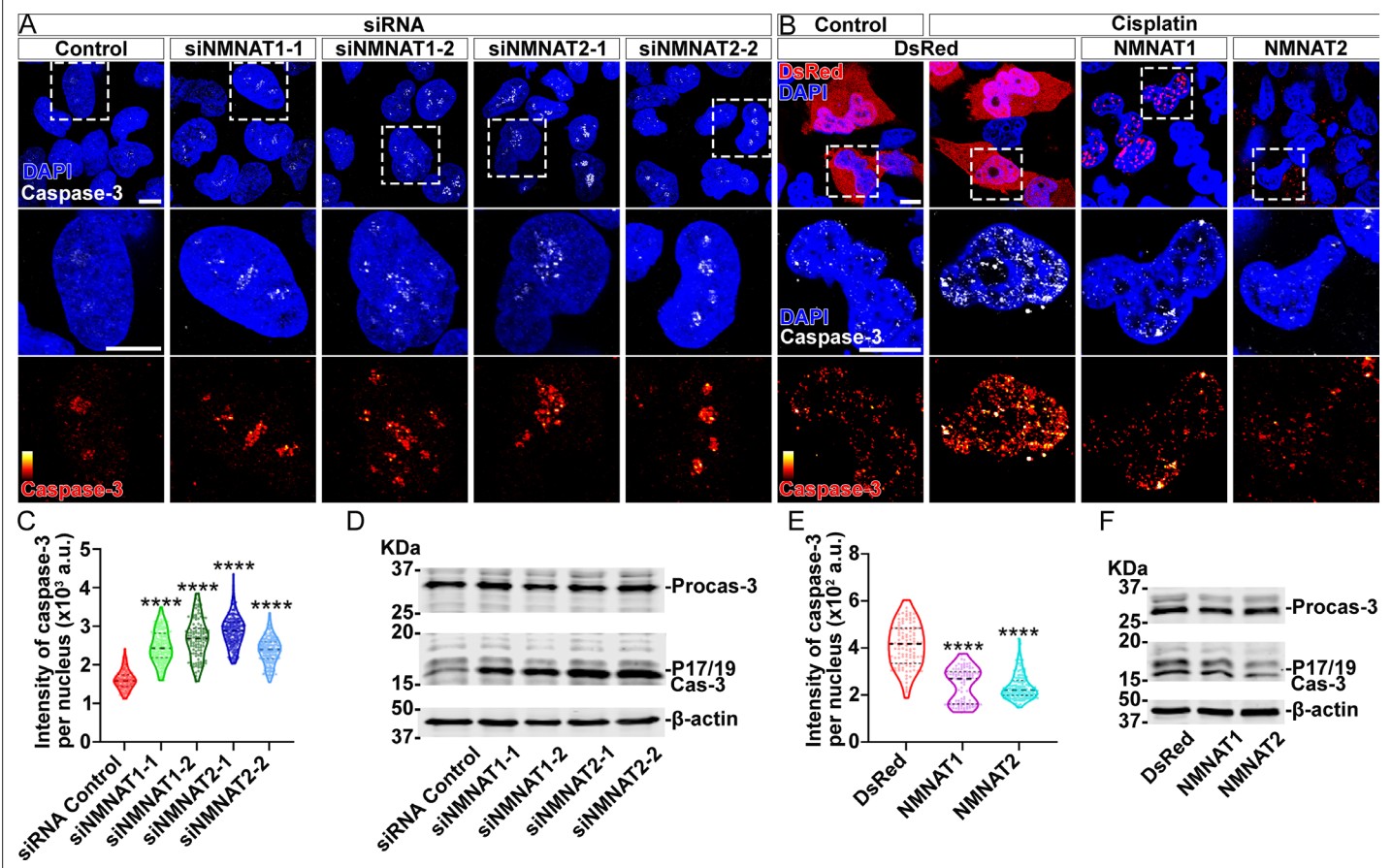

**Figure 5.** NMNAT decreases caspase-3 activation in human glioma cells. (**A**) T98G cells were transfected with siNMNAT1 or siNMNAT2 and stained with DAPI (blue) and caspase-3 (white). (**B**) T98G cells were transfected with DsRed (red), DsRed-NMNAT1 (red), or DsRed-NMNAT2 (red), treated with cisplatin 8 hr after transfection, and stained with DAPI (blue) and caspase-3 (gray). The second and third rows are the high magnification of the boxed areas in the first row. In the third row, the intensity of caspase-3 is indicated by a heatmap (0–4095). Scale bars, 10 μm. (**C**) Quantification of nuclear caspase-3 intensity in (**A**). Data are presented as median ± quartiles, n≥100. Significance level was established by one-way ANOVA post hoc Bonferroni test. (**E**) Quantification of nuclear caspase-3 intensity in (**B**). Data are presented as median ± quartiles, n≥100. Significance level was established by one-way ANOVA post hoc Bonferroni test. (**D, F**) Proteins were extracted from T98G cells transfected with siRNA (**D**), plasmids and treated with cisplatin for 8 hr (**F**) for Western blot analysis. P17/19 was considered as cleaved caspase-3. β-actin was used as an internal control. ****p≤0.0001. NMNAT, nicotinamide mononucleotide adenylyltransferase.

The online version of this article includes the following source data and figure supplement(s) for figure 5:

**Source data 1.** NMNAT decreased caspase-3 activation in human glioma cells.

**Figure supplement 1.** Membrane of Western blot analysis.

**Figure supplement 2.** Cleaved caspase-3 is reduced after NMNAT overexpression.

**Figure supplement 2—source data 1.** Cleaved Ccaspase-3 was reduced after NMNAT overexpression.

the presence of a trimeric p53/ NMNAT/PARP1 complex, and a potential role of NMNAT in promoting the trimeric complex formation.

To confirm and extend the biochemical analysis, we carried out immunofluorescent colocalization studies of T98G glioma cells expressing NMNAT1 and detected the colocalization of NMNAT1 with p53 (*Figure 9E1*) as well as of NMNAT1 with PARP1 (*Figure 9G1*). Consistent with Western blot analysis (*Figure 9C*), p53 protein level is not altered by NMNAT expression as p53 immunofluorescence intensity was similar between NMNAT1 expression cells and neighboring untransfected cells, or DsRed expressing control cells (*Figure 9D2* and quantified in *Figure 9H*). Interestingly, the distribution of p53 changed from diffuse to clustered in NMNAT1-positive hotspots, as visualized by a fluorescence surface plot in *Figure 9E2'*. Similarly, PARP1 protein also clustered in NMNAT1-positive hotspots (*Figure 9G2 and G2'*), suggesting the close proximity of NMNAT, p53, and PARP1. In addition, in

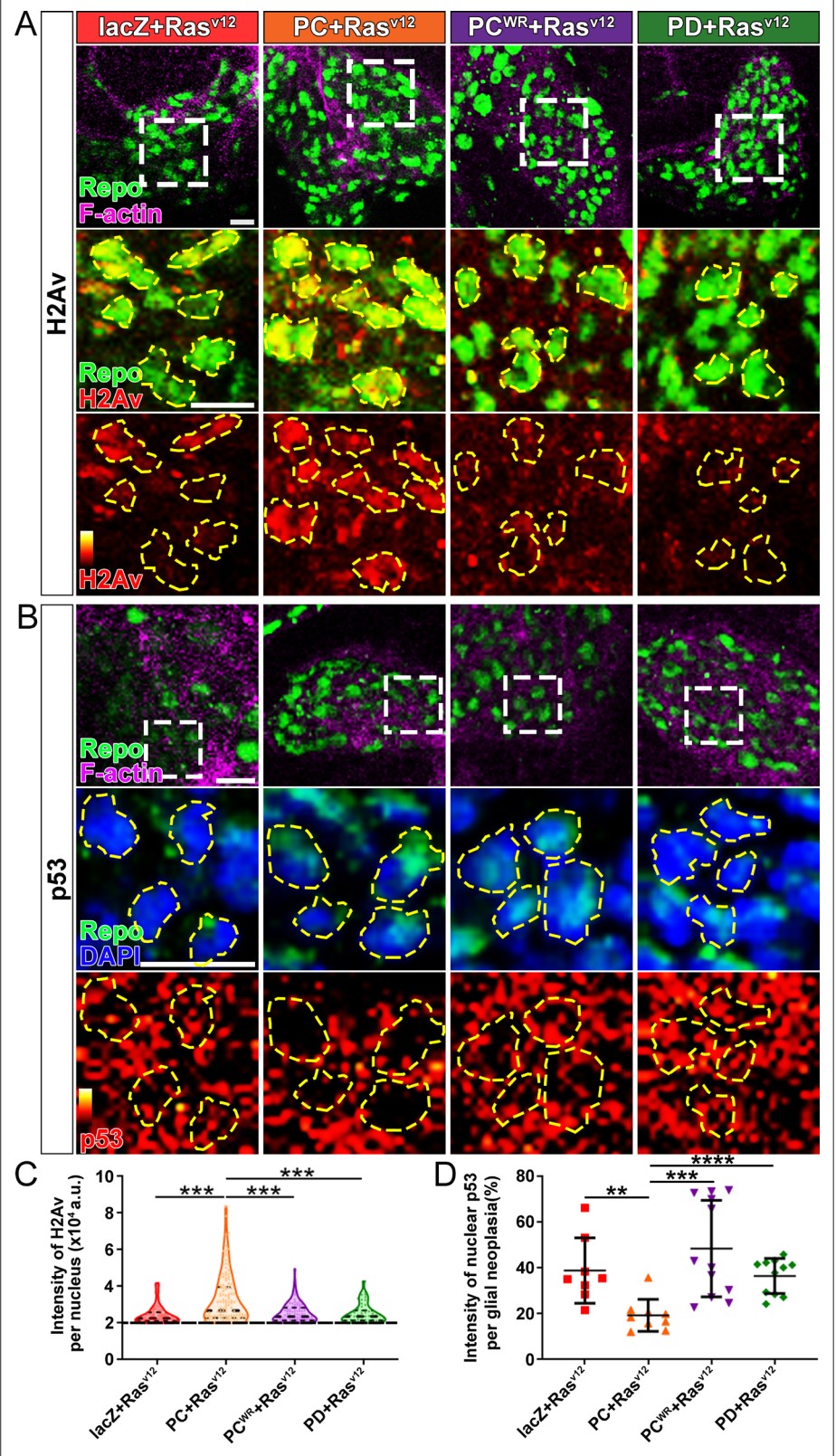

**Figure 6.** Nmnat-PC inhibits DNA damage-induced p53 activation in glial neoplasia. (**A**) Glial neoplasia from flies expressing lacZ, PC, PC^WR^, or PD were stained with H2Av (red), Repo (green), and F-actin (magenta). The second and third rows are high magnification of the boxed areas in the first row. In the third row, the intensity of H2Av is indicated by a heatmap (0–4095). (**B**) Glial neoplasia from flies expressing lacZ, PC, PC^WR^, or PD were stained with

*Figure 6 continued on next page*

*Figure 6 continued*

p53, Repo (green), F-actin (magenta), and DAPI (blue). The second and third rows are high magnification of the boxed areas in the first row. In the third row, the intensity of p53 is indicated by a heatmap (0–4095). Yellow dashed lines indicate the nuclear areas. Scale bars, 10 μm. (**C**) Quantification of H2Av intensity in Repo-positive cells. The black dashed line indicates the threshold. According to the lacZ group, value 20,000 is set as the threshold. Data are presented as median ± quartiles, n≥3. Significance level was established by one-way ANOVA post hoc Bonferroni test. (**D**) Quantification of nuclear p53 intensity. Data are presented as mean ± s.d., n≥3. Significance level was established by t-test. **p≤0.01; ***p≤0.001; ****p≤0.0001.

NMNAT-expressing cells, PARP1 levels exhibit a small but significant upregulation (*Figure 9I*). Collectively, these results suggest that NMNAT interacts with PARP1 and promotes PARylation of PARP1-targeting proteins, including p53, through increasing the local $NAD^+$ availability.

In addition to PARylation, p53 is modified by another $NAD^+$-dependent posttranslational modification (PTM), deacetylation. p53 is acetylated by p300/CBP and deacetylated by SIRTs family of $NAD^+$-dependent deacetylases (*Vaziri et al., 2001*). SIRT1 is the major deacetylase regulating p53 activity through deacetylation of p53 at K382, and hence inhibiting the p53-mediated apoptosis pathway (*Cheng et al., 2003*). NMNAT1 has been reported to interact with SIRT1 directly (*Zhang et al., 2009*). We determined the level of acetyl-p53 in cell extracts and through immunoprecipitating by anti-p53 antibodies from T98G glioma cells with or without NMNAT overexpression, and then probing for acetyl-p53 at K382. Interestingly, with NMNAT1 or NMNAT2 overexpression, acetyl-p53 was specifically reduced while total p53 levels remained the same (*Figure 10A–C*), although a stable complex of p53 and SIRT1 was not detected. It is interesting to note that endogenous SIRT1 expression was upregulated in NMNAT overexpressing cells (*Figure 10D*), suggesting a potential coregulation of NMNAT and SIRT1. Notably, similar results were observed in U87MG cells (*Figure 10—figure supplement 1*), suggesting a common effect of NMNAT on p53 modification. Collectively, these results show that NMNAT upregulation promotes the $NAD^+$-dependent deacetylation of p53 and specifically reduces the pool of acetyl-p53.

To further examine the disease-relevant role of NMNAT in glioma growth, we analyzed patient data from the Cancer Genome Atlas (TCGA) to determine how NMNAT expression levels affect survival in glioma and glioblastoma, using the gene expression profiling interactive platform, GEPIA (http://gepia.cancer- pku.cn/). A strong negative correlation between NMNAT1 expression and survival can be seen in patients with brain lower grade glioma (LGG), with elevated NMNAT1 expression significantly associated with a lower disease-free survival rate, both when comparing survival in the median high-low tumor expression patient groups (*Figure 11A*) and in the highest and lowest 10% expression groups (*Figure 11B*). In the aggressive form of glioma, glioblastoma multiforme (GBM), from which the T98G and U87MG cell lines are derived, high NMNAT1-expressing tumors (top 10%) again showed a significant correlation with more aggressive disease and poorer outcome (*Figure 11C*). However, NMNAT2 GBM expression levels did not correlate with patient survival (*Figure 11D*). These brain glioma patient data set results indicate the strong correlation between high NMNAT1 expression with lower survival and poorer clinical outcome. As both PARylation and deacetylation modifications of p53 have been reported to inactivate p53-mediated function and activity (*Juan et al., 2000*; *Luo et al., 2000*; *Malanga et al., 1998*; *Simbulan-Rosenthal et al., 1999*), collectively, our results suggest a model where NMNAT promote glioma growth through facilitating $NAD^+$-dependent PTMs of p53 to ameliorate apoptosis (*Figure 11E*).

## Discussion

In this study, we identified a critical role of NMNAT in promoting glioma cell proliferation and growth in a model of *Drosophila* glial neoplasia and human glioma cell lines. We found that NMNAT promotes glioma growth by allowing a higher tolerance to DNA damage and inhibiting p53/caspase-mediated apoptosis. Mechanistically, upregulation of enzymatically active NMNAT promotes the $NAD^+$-dependent PTMs of p53. Specifically, we detected upregulation of protein PARylation and the presence of a p53-NMNAT-PARP1 trimeric complex as well as decreased acetylation of p53 accompanied with increased SIRT1. Our findings support a tumorigenesis model where NMNAT proteins promote glioma growth through regulating $NAD^+$-dependent PTM of p53, and driving cellular pools

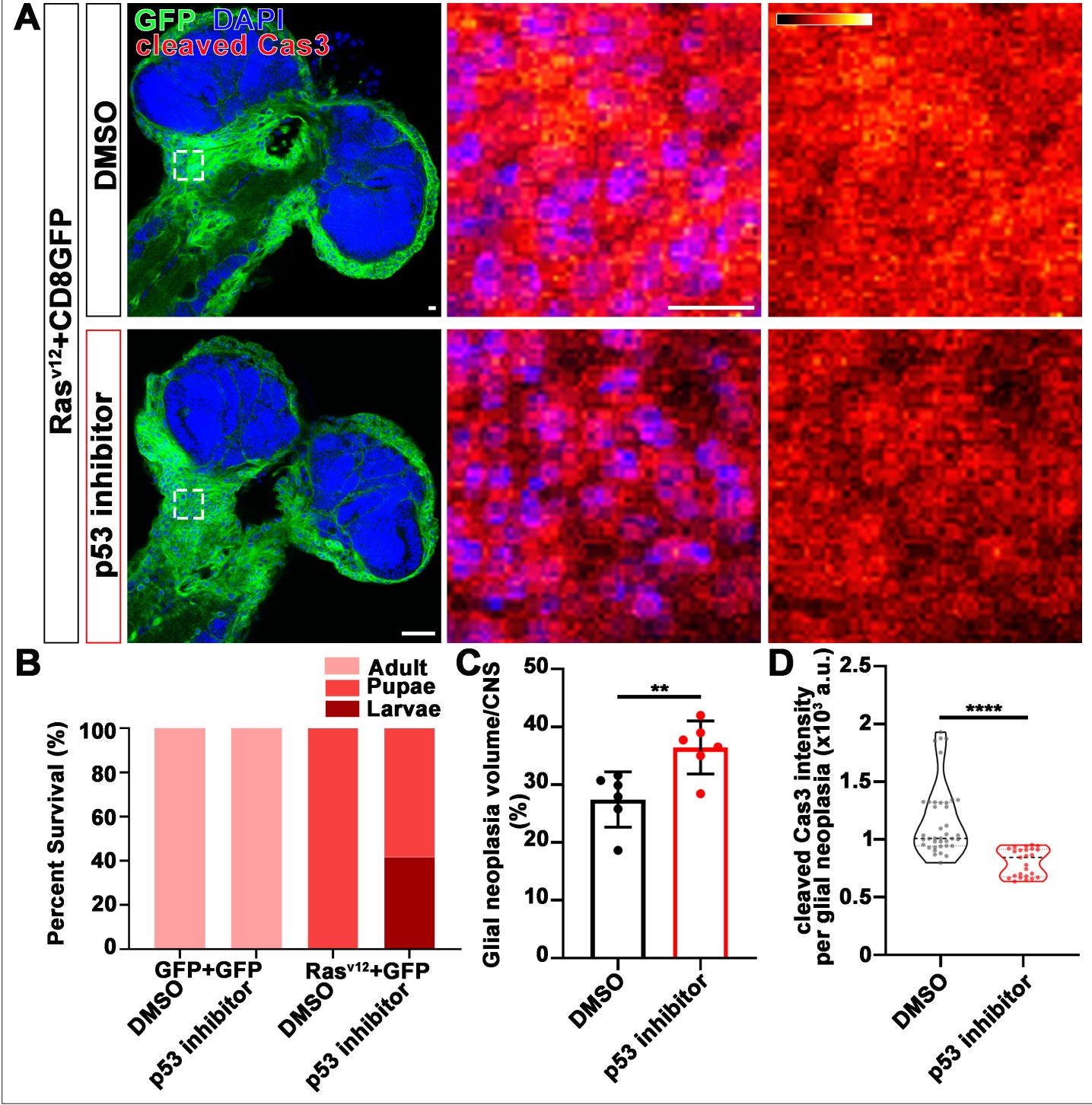

**Figure 7.** p53 inhibitor increases glial neoplasia volume in CNS and larvae lethality. (**A**) Flies expressing Ras$^{v12}$ and CD8GFP were treated with DMSO or p53 inhibitor, respectively, stained with cleaved caspase-3 (red) and DAPI (blue). The first column is the whole CNS of flies. White dashed lines indicate the glial neoplasia areas. The second and third columns are high magnification of the boxed white areas in the first row. The intensity of cleaved caspase-3 is indicated by a heatmap (0–4095). Scale bars, 10 µm. (**B**) Survival rate of flies. (**C**) Quantification of ratio of glial neoplasia volumes in CNS. Data are presented as mean ± s.d., n≥3. Significance level was established by one-way ANOVA post hoc Bonferroni test. (**D**) Quantification of cleaved caspase-3 intensity. Data are presented as median ± quartiles, n≥3. Significance level was established by one-way ANOVA post hoc Bonferroni test. **p≤0.01; ****p≤0.0001. CNS, central nervous system.

of p53 toward PARylated-p53 (inactive p53) and away from acetyl-p53 (active p53) to ameliorate

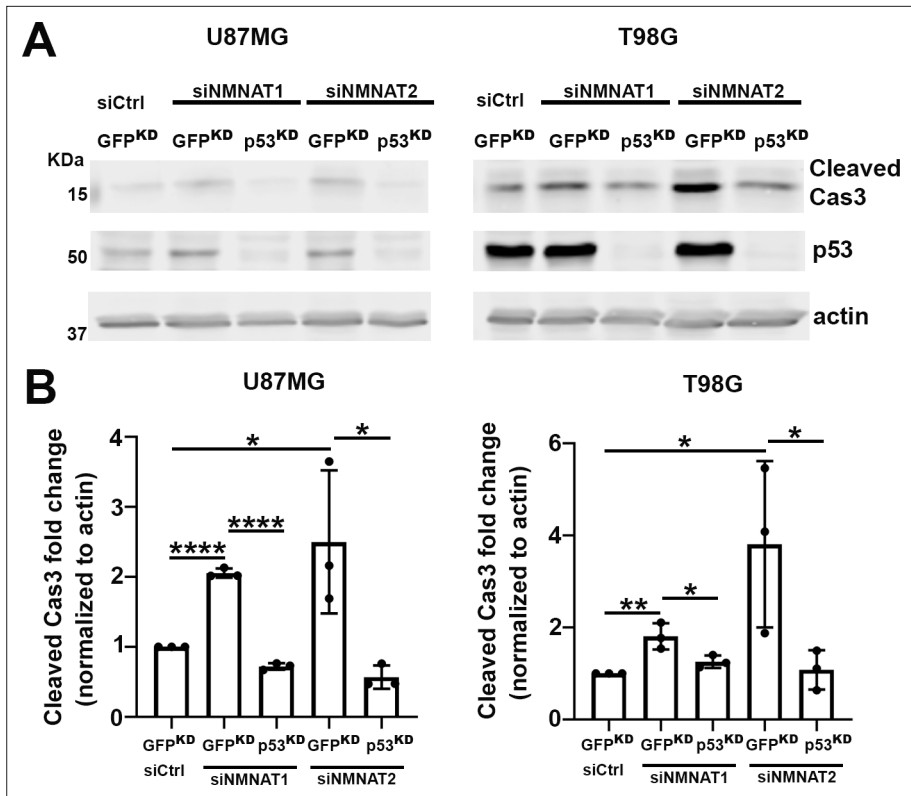

**Figure 8.** p53 depletion rescues NMNAT knockdown induced caspase-3 activation in glioma. (**A**) Proteins were extracted from U87MG and T98G cells transfected with siNMNAT1/2 after knockdown of p53 or GFP by shRNA lentivirus transduction for Western blot analysis. β-actin was used as an internal control. (**B**) Quantification of cleaved caspase-3 in Western blot analysis. Data are presented as mean ± s.d., n≥4. Significance level was established by t-test. *p≤0.05; **p≤0.01; ****p≤0.0001. NMNAT, nicotinamide mononucleotide adenylyltransferase.

The online version of this article includes the following source data and figure supplement(s) for figure 8:

**Figure supplement 1.** p53 depletion rescues NMNAT knockdown induced caspase-3 activation in glioma.

**Figure supplement 1—source data 1.** p53 depletion rescued NMNAT knockdown-induced caspase-3 activation in glioma.

**Source data 1.** p53 depletion rescued NMNAT knockdown-induced caspase-3 activation in glioma.

DNA damage-triggered cell death under oncogenic stresses associated with tumor development (*Figure 11E*).

## The advantages and potential of an in vivo *Drosophila* glial neoplasia

We adapted a glial neoplasia model in *Drosophila* using the *UAS-Ras85D$^{v12}$* and repo-GAL4 driver system that induces overgrowth of glial cells to mimic glial neoplasia formation (*Read et al., 2009*). Although RAS alterations in human glioma occur at a lower frequency than some other higher frequency driver alterations (*Brennan et al., 2013*), our rationale for using mutant RAS overexpressing model in *Drosophila* was to study the effects of NMNAT on the broader common (rather than Ras-specific) processes underlying tumorigenic development in a validated glioma model in *Drosophila*. It will be an important future direction to establish *Drosophila* models using other high-frequency glioma drivers. Since all *Drosophila* glia express Repo, we can easily monitor the formation of *Ras$^{v12}$*-driven glial neoplasia in the brain by GFP reporter, Repo, and F-actin labeling. In fluorescence imaging, normal brains typically have two to three layers of Repo-positive cells visible in each section (*Figure 1B*). Therefore, any tissue mass consisting of more than three layers of glia would be atypical and potentially tumor-like. We analyzed glial neoplasia with three key criteria: cell type (Repo-positive), cell number (more than three layers with at least 400), and tissue size (volume of at least 12.4×10³ μm³). Combined with our high-resolution imaging capability, these criteria allow us to distinguish tumor

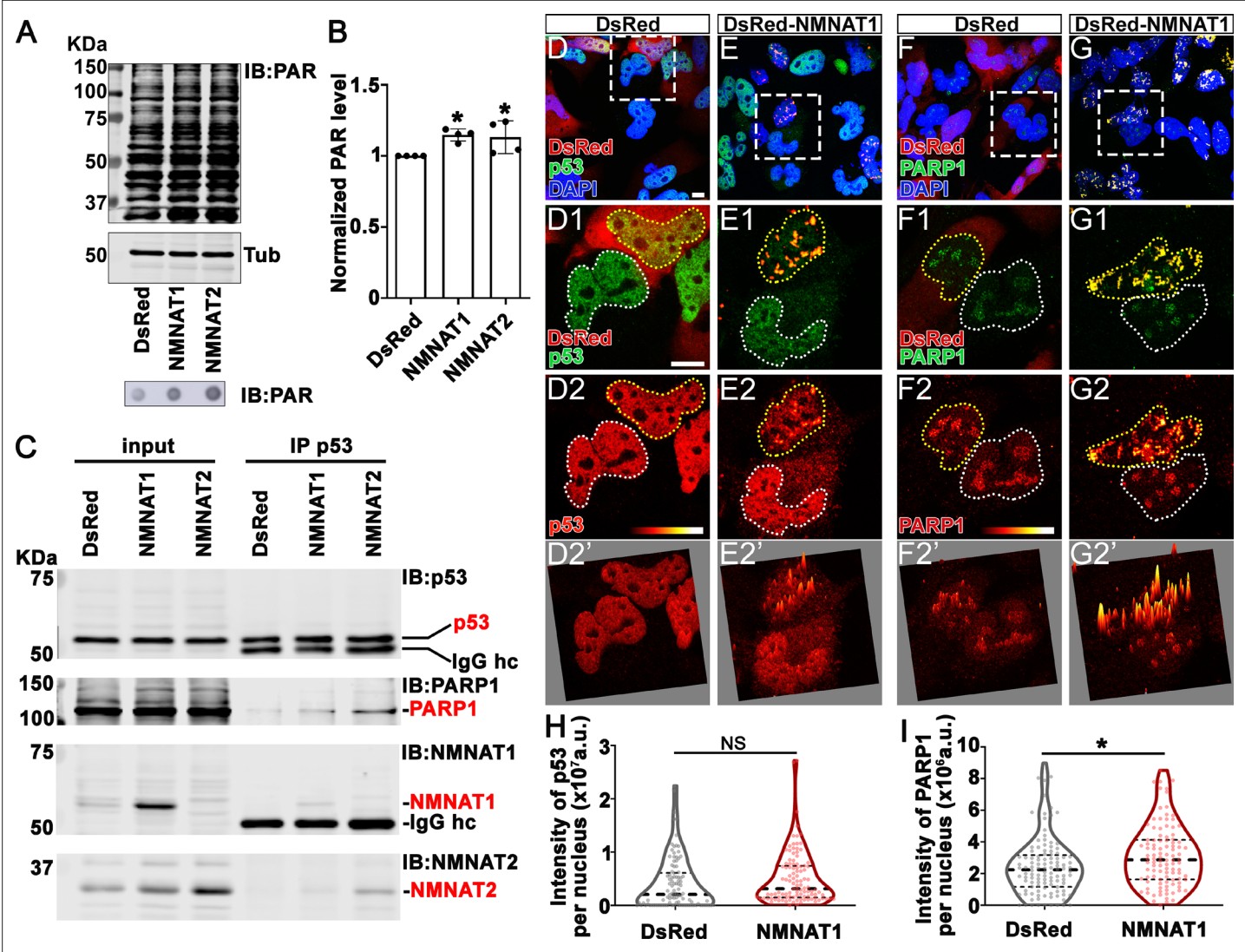

**Figure 9.** NMNAT interacts with p53 and PARP1 and upregulates PARylation. (**A**) Proteins were extracted from T98G cells transfected with plasmids for Western blot and dot blot analyses using anti-PAR antibody. (**B**) Quantification of PAR in Western blot analysis. Data are presented as mean ± s.d., n=4. Significance level was established by t-test. (**C**) Protein samples extracted from T98G cells transfected with DsRed, DsRed-NMNAT1, or NMNAT2 were immunoprecipitated (IP) with a p53 antibody and subjected to immunoblot (IB) analysis for p53, PARP1, NMNAT1, and NMNAT2. (**D–G**) T98G cells transfected with DsRed or DsRed-NMNAT1 were stained for DAPI (blue), p53 (green), or PARP1 (green). The second to the fourth rows are high magnification of the boxed area in the first row. The intensity (0–4095) of p53 or PARP is indicated in a heatmap (**D2–G2**) or surface plot (**D2'–G2'**). Scale bars, 10 μm. (**H**) Quantification of nuclear p53. Data are presented as median ± quartiles, n≥100. Significance level was established by one-way ANOVA post hoc Bonferroni test. (**I**) Quantification of PARP1 intensity. Data are presented as median ± quartiles, n≥100. Significance level was established by one-way ANOVA post hoc Bonferroni test. *p≤0.05. NS, not significant. NMNAT, nicotinamide mononucleotide adenylyltransferase.

The online version of this article includes the following source data and figure supplement(s) for figure 9:

**Source data 1.** NMNAT interacts with p53 and PARP1 and upregulates PARylation.

**Figure supplement 1.** PARylation is reduced after NMNAT knockdown.

**Figure supplement 2.** NMNAT interacts with p53 in U87MG.

**Figure supplement 2—source data 1.** NMNAT interacts with p53 in U87MG.

from non-glial neoplasia tissue with high confidence and to analyze glial neoplasia in the most robust and reproducible manner. In $Ras^{v12}$ expressing flies, we observed glial neoplasia occurred extensively in the brain and VNC.

In addition to the morphological phenotypes, we found that glial neoplasia reduced the animal survival rate. Specifically, the total volume of glial neoplasia tissue is positively correlated with the

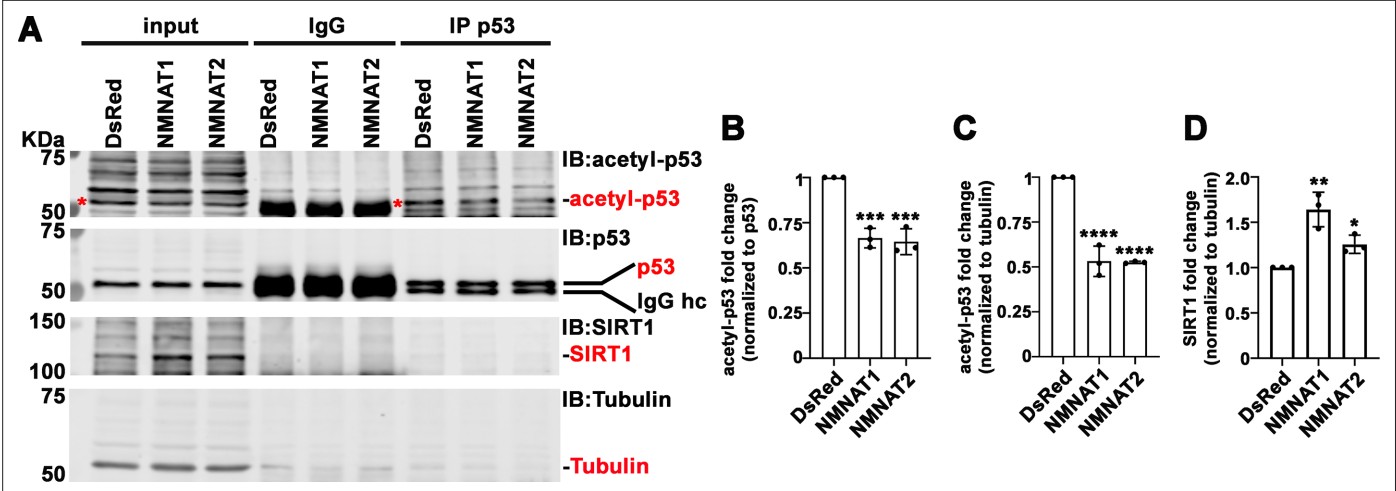

**Figure 10.** NMNAT upregulates SIRT1 and reduces acetylation of p53. (**A**) Protein samples extracted from T98G cells transfected with DsRed, DsRed-NMNAT1, or NMNAT2 were immunoprecipitated (IP) with a p53 antibody and probed for acetyl-p53 and SIRT1. Red asterisks (*) indicate the acetyle-p53 bands in input and IP-ed fractions. (**B, C**) Quantification of acetyl-p53 in p53-immunoprecipitated fraction (**B**) and input fraction (**C**). (**D**) Quantification of SIRT1. Data are presented as mean ± s.d., n=3. Significance level was established by one-way ANOVA post hoc Bonferroni test. *p≤0.05; **p≤0.01; ***p≤0.001. NMNAT, nicotinamide mononucleotide. adenylyltransferase.

The online version of this article includes the following source data and figure supplement(s) for figure 10:

**Source data 1.** NMNAT upregulates SIRT1 and reduces acetylation of p53.

**Source data 2.** NMNAT upregulates SIRT1 and reduces acetylation of p53.

**Figure supplement 1.** Acetyl-p53 is reduced after NMNAT overexpression in U87MG.

**Figure supplement 1—source data 1.** Acetyl-p53 was reduced in U87MG cells under NMNAT overexpression conditions.

severity of reduced animal survival rate. Such correlation allows the use of high-resolution in vivo morphological imaging as a strong predictor of pathological outcome and a powerful tool to identify genetic modulators of tumorigenesis as we have done in this study, and potential pharmacological modulators for cancer therapy in the future.

## NMNAT-mediated NAD⁺ biosynthesis promotes glioma growth

Our results show that NMNAT expression promotes glioma growth but is likely dispensable for its initiation, as NMNAT overexpression alone did not trigger tumorigenesis. Our results showed that the enzymatic function of NMNAT is required for glioma growth. This finding is not surprising given the fundamental role of NAD⁺ as a signaling cofactor that regulates cancer metabolism through its coenzymatic function in the redox reactions underlying essential bioenergetic pathways, including glycolysis, the TCA cycle, and oxidative phosphorylation (*Hanahan and Weinberg, 2000*). While NAMPT is the rate-limiting enzyme, NMNAT is downstream of NAMPT and directly regulates the level of NAD⁺ by catalyzing the reversible reaction of NAD⁺ synthesis. The direction of the reaction, forward (NAD⁺ production) or reverse (NAD⁺ breakdown), is dependent upon the availability of subtracts. Therefore, NMNAT functions as a cellular metabolic sensor and maintains the homeostasis of NAD⁺ pools.

NAD⁺ is highly compartmentalized, with each subcellular NAD⁺ pool differentially regulated and preferentially involved in distinct NAD⁺-dependent signaling or metabolic events (*Zhu et al., 2019*). Compartment-localized NMNAT isoforms contribute to the maintenance of subcellular NAD⁺ pools. In mammals, NMNAT1 is nuclear and NMNAT2 is cytoplasm-localized (*Berger et al., 2005*). Our analysis of the public glioma cancer data set GEPIA identified a strong negative correlation between NMNAT1 expression and disease-free survival in patients with brain LGG as well as the progressive GBM. These findings have several implications. First, the critical requirement for nuclear NAD⁺-consuming events in tumor growth demands a constant supply of nuclear NAD⁺ pool by nuclear-localized NMNAT. Indeed, as our results show, NAD⁺-dependent PARylation and deacetylation of proteins including p53 underlies the mechanism of tumorigenesis. Second, the difference in the tumor-promoting effects of nuclear vs. cytoplasmic NMNAT isoforms may inform cellular metabolic needs and genotoxic load.

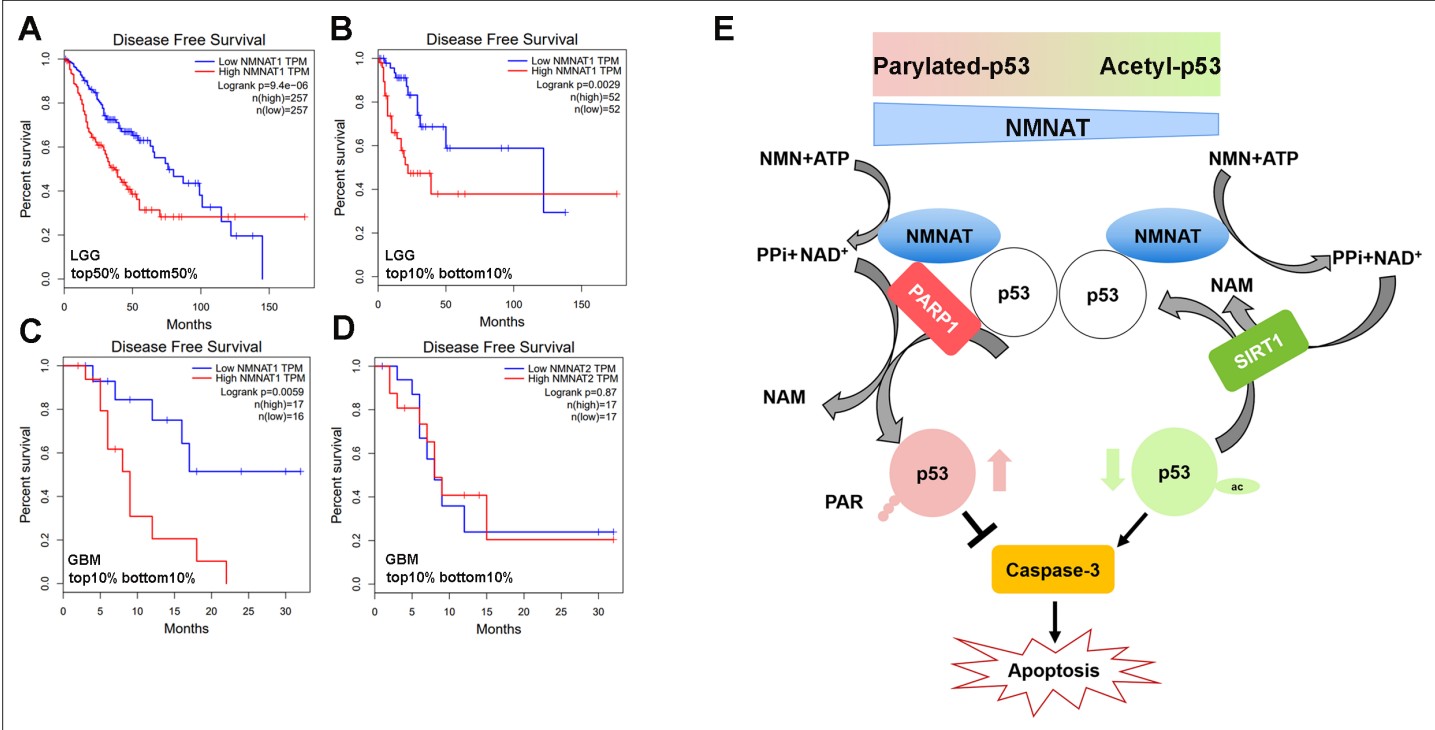

**Figure 11.** Correlation of tumor NMNAT expression levels with glioma progression. (**A, B**) LGG (lower grade glioma) data set (GEPIA). Survival curves for patients with tumors expressing median (top 50% vs. bottom 50%) NMNAT1 levels (**A**), or top 10% versus bottom 10% levels (**B**), with associated log-rank p-values shown. Highly significant correlation of high NMNAT1 expression levels with poor patient survival and more aggressive disease. (**C, D**) GBM (glioblastoma multiform) data set (GEPIA). Survival curves for patients are shown, with tumors expressing top 10% versus bottom 10% NMNAT1 levels (**C**) and NMNAT2 (**D**) levels as well as associated log-rank p-values. High NMNAT1-expressing tumors have extremely poor relapse-free rates compared to low NMNAT1-expressing tumors. In contrast, no significant difference in survival is detected for NMNAT2 expression. (**E**) Model for NMNAT in glioma. In glioma cells, PAPR1 inhibits p53 activity by NAD$^+$ dependent-poly(ADP-ribosyl)ation of p53 during DNA damage repair. NMNAT overexpression replenishes the NAD$^+$ pool to promote poly(ADP-ribosyl)ation and deacetylation of p53, suppressing p53 induced apoptosis, thereby leading to glioma growth. NMNAT, nicotinamide mononucleotide adenylyltransferase.

The online version of this article includes the following figure supplement(s) for figure 11:

**Figure supplement 1.** Summary of NMNAT1 and NMNAT2 alteration frequency in cancer types.

**Figure supplement 2.** Summary of NMNAT1 and NMNAT2 alteration frequency in cancer types.

Interestingly, the CBio portal databases show NMNAT1 and NMNAT2 genes appear to be amplified in distinct cancer types (*Figure 11—figure supplements 1 and 2*). Future work is required to identify the specific roles of NMNAT1 and NMNAT2 in different cancer types.

During the submission of this manuscript, two groups reported the distinct roles of NMNAT1 and NMNAT2 in acute myeloid leukemia and ovarian cancer respectively (*Challa et al., 2021*; *Shi et al., 2021*). *Shi et al., 2021* showed that NMNAT1-mediated NAD$^+$ metabolism regulates p53 acetylation and enables acute myeloid leukemia (AML) to evade apoptosis (*Shi et al., 2021*). *Challa et al., 2021* showed that NMNAT2-mediated cytosolic NAD$^+$ synthesis regulates ribosome ADP-ribosylation to maintain protein homeostasis in ovarian cancer (*Challa et al., 2021*). It is important to note that in mammalian cells, nuclear and cytoplasmic NMNATs can regulate each other's activity, likely through feedback from dynamic pool of substrates NMN and ATP, as overexpressing cytoplasmic NMNAT may exhaust the supply of NMN therefore repress nuclear NAD$^+$ synthesis (*Ryu et al., 2018*). Consequently, altering the nuclear NAD$^+$ pool may regulate gene transcription and influence cell differentiation or proliferation state (*Ryu et al., 2018*). Our observation of the specific upregulation of endogenous nuclear NMNAT upon oncogenic RAS-expression further supports the hypothesis that nuclear and cytoplasmic NMNAT react differently in stress conditions and likely be important in different stages of tumor growth.

## NAD+-mediated posttranslational modifications of P53: a balancing act

PARylation, phosphorylation, acetylation, and ubiquitination are PTMs that have been shown to regulate the stability and activity of p53 (*Bode and Dong, 2004*). Among the most common PTMs of p53, PARylation and acetylation are both NAD+-consuming processes mediated by NAD+-dependent enzymes, PARPs, and SIRTs (*Lee et al., 2012*; *Vaziri et al., 2001*). When PARP1 activity is induced in the DNA damage response process, extensive protein PARylation occurs and many proteins including p53 and DNA repair machinery components are PARylated (*Amé et al., 2004*; *Fischbach et al., 2018*). Numerous studies have shown that PARylation of p53 may inhibit p53-mediated function, including cell cycle arrest and apoptosis (*Kanai et al., 2007*; *Simbulan-Rosenthal et al., 1999*; *Simbulan-Rosenthal et al., 2001*). With abundant NAD+ supply, PARylation is an efficient way to repair DNA damage and ensure cell survival; whereas under conditions of insufficient NAD+ supply, apoptosis is induced (*Herceg and Wang, 2001*). In response to DNA damage, the activity of p53 is also modulated by acetylation. Acetyl-p53 is resistant to degradation by ubiquitination and has higher stability, and therefore can exert longer effects of growth arrest, senescence, and apoptosis (*Li et al., 2002*). NAD+-dependent PARylation and acetylation have the opposite effects on p53 activity, where PARylation inhibits p53 activity and acetylation prolongs p53 activity. NAD+ thus plays a critical role in balancing the pro-apoptotic activity of p53. NMNAT1 regulates functions of NAD+-dependent enzymes such as SIRT1 and PARP1 (*Zhang et al., 2009*; *Zhang et al., 2012*). Interestingly, our results identified a trimeric complex of NMNAT-PARP1-p53 and increased PARP1 and SIRT1, which supports the model that NMNAT recruits NAD+-utilizing enzymes, including PARP1 and SIRT1, together with protein substrates, and locally supply NAD+ for NAD+-dependent protein modification. Such an NMNAT-PTM modifying enzyme-protein substrate trimeric protein complex will not only sustain the local supply of NAD+ but also facilitate and expedite the modification process.

It is important to note that p53 is not the only target for PARylation and deacetylation regulation. The role of NMNAT in PARylation of other target proteins has also been indicated. For example, it has been shown that decreased NMNAT1 expression caused nuclear NAD+ deficiency and subsequently reduced PARylation of multifunctional nuclear protein CCCTC-binding factor, leading to epigenetic silencing of tumor suppressor genes (*Henderson et al., 2017*). As noted above, the most recent study showed increased NMNAT2 mediated cytosolic NAD+ synthesis activity supports mono(ADP-ribosyl) ation (MARylation) through PARP-16 in ribosome (*Challa et al., 2021*). These reports together with our findings support a specific role of NAD+ in modulating tumorigenesis through regulating PTMs, including PARylation and deacetylation. Our findings in both in vivo and in vitro models highlight NMNAT's roles in promoting glioma development. Specifically, the direct interaction we identified among p53, NMNAT, and PARP1 has important implications regarding the utility of NMNAT as a potential target for glioma therapy. Because the protein-protein interaction interface of NMNAT-PARP1-p53 could provide allosteric targeting of NMNAT, in addition to its enzyme pocket, this may open new possibilities for alternative inhibitors of NAD+-dependent pathway with less toxicity.

It should be noted that although T98G cells carry a gain-of-function p53 mutation (M237I) in the DNA binding domain, this mutation does not affect the sites of PARylation and acetylation (*Yamamoto and Iwakuma, 2018*; *Yi et al., 2013*). Moreover, prior studies support the ability of cells harboring this p53 mutant to undergo apoptosis, which can be abrogated by p53 inhibition (*Enns et al., 2004*). Our findings that NMNAT similarly affects p53 modification in either wild-type (U87MG) or mutant p53 (T98G) cells suggest NAD+-dependent PARylation or deacetylation of p53 is independent of the p53 [M237I] mutation. Indeed recent studies have shown that mutant p53 proteins retain the ability to induce apoptosis despite losing tumor-suppressive transactivation functionality (*Timofeev et al., 2019*). Further studies will be required to fully understand the effects of NMNAT on p53 transcription factor function.

In conclusion, our studies have identified NMNAT as an NAD+ synthase that plays an essential role in regulating the function and activation of p53 during DNA damage-induced apoptosis in glioma cells. These results support the development of specific NMNAT inhibitors as potentially efficacious therapeutic agents in cancers with upregulated NMNAT levels.

## Materials and methods

### Fly stocks and culture

Flies were maintained at 25°C room temperature with standard medium. The following lines were used in this study obtained from the Bloomington *Drosophila* Stock Center: (1) The driver used in all experiments: *repo-GAL4*; (2) *UAS-Ras$^{v12}$* (II); (3) *UAS-Ras$^{v12}$* (III); (4) *UAS-Nmnat RNAi* (III); (5) *UAS-p35*; (6) *UAS-Diap1*; (7) *UAS-Dronc* RNAi; and (8) *UAS-DCP1 RNAi*. UAS-*Drosophila melanogaster* Nmnat (*UAS-PC, UAS-PC$^{WR}$, UAS-PD*) were generated in the laboratory.

### Fly treatment

Larvae were collected and treated with 100 µM of Pifithrin-α (Sigma-Aldrich, P4359) with standard medium at 25°C room temperature.

### Human glioma cell line culture and treatment

T98G and U87MG (human glioma cells) cell lines were purchased from the American Type Culture Collection (ATCC; CRL-1609). SVG p12 cell line was from Dr. Michal Toborek (University of Miami). Cells were maintained in Eagle's Minimum Essential Medium (EMEM; Sigma-Aldrich, M0325) supplemented with 10% fetal bovine serum (FBS; ATCC, 30–2020). Cells were cultured at 37°C, 5% $CO_2$. To induce apoptosis, cells were treated with 50 µM of cisplatin for 8 hr (Sigma-Aldrich, 232120).

### Antibodies

The following commercially available antibodies were used: anti-Repo (1:250, DSHB, 8D12), anti-Caspase-3 (1:250 for Immunocytochemistry of fly brain, 1:1000 for Western blot analysis, Cell Signaling Technology, 9665), anti-Cleaved Caspase-3 (1:1000, Santa Cruz, 9661), anti-H2AvD (1:50, Rockland, 600-401-914), anti-p53(E-5) (1:50, Santa Cruz, sc-74573), p53(DO-1) (1:1000, Santa Cruz, sc-126), anti-*Drosophila* Nmnat (1:3000), anti-NMNAT1 (1:1000, Abcam, ab45548), anti-NMNAT1 (1:1000, Santa Cruz, 271557), anti-NMNAT2 (1:500, Abcam, ab56980), anti-PARP1 (1:1000, Santa Cruz, sc-8007), anti-pADPr (1:1000, Santa Cruz, sc-56198), anti-SIRT1 (1:1000, Cell Signaling Technology, 2492), anti-acetyl-p53 (1:1000, Cell Signaling Technology, 2525), anti-β-actin (1:10,000, Sigma-Aldrich, A1978), and anti-tubulin (1:300, Abcam, ab15246). The secondary antibodies conjugated to Alexa 488/546/647 (1:250, Invitrogen), or near-infrared (IR) dye 700/800 (1:5000, LI-COR Biosciences). HRP-anti-mouse and HRP-anti-rabbit (1:5000, Thermo Fisher Scientific).

### Plasmid construction

Four recombinant plasmids were generated for this study: pDsRed, pDsRed-NMNAT1, pDsRed-NMNAT2, NMNAT1, and NMNAT2.

### RNA interference

Small interference RNA sequences targeting human NMNAT were purchased (GenePharma). The siRNA sequences were listed in Supplementary (*Figure 3—figure supplement 2*).

### shRNA knockdown

Stable p53 knockdown was performed using the pLKO lentiviral shRNA system. Lentiviral supernatant production was carried out in HEK 293T cells and transduction of target U87MG and T98G cells with either the shp53 or the control shGFP supernatant was performed as described previously (*Rai et al., 2009*). The following validated shRNA target sequences (*Burton et al., 2013*); (*Patel et al., 2015*) were used:

> shGFP: 5'-GCAAGCTGACCCTGAAGTTCA-3'
> shp53: 5'-GACTCCAGTGGTAATCTACTT-3'

Transduced cells were selected in 2.5 µg/ml puromycin-containing culture media for a minimum period of 5–7 days (corresponding to the time taken for untransduced cells to die completely in selection media).

## Real-time RT-PCR

The total RNA was extracted by TRIzol reagent (Invitrogen) from T98G cells according to the manufacturer's protocol. cDNA was synthesized from RNA with a cDNA Reverse Transcription Kit (Applied Biosystems). RNA was performed using a Real-Time System and SYBR Green Kit (Applied Biosystems). Relative gene expression was compared to actin as an internal control. The primers used in detection were listed in the Supplementary (*Figure 3—figure supplement 2*).

## Cells transfection

Cells for transfection were seeded in a six-well culture vessel (VWR) containing EMEM media with 10% FBS for 24 hr. Plasmids or siRNA were transfected with transfection reagent (jetPRIME). Gene expression was measured by Western blot analysis and real-time qPCR after cells were transfected at 48 hr.

## Immunocytochemistry of cells

Cells were grown on 22 mm glass coverslips (VWR). After treatment, cells were rinsed three times with phosphate-buffered saline (PBS), fixed for 15 min in 4% paraformaldehyde, washed three times with PBS, and permeabilized with 0.4% Triton X-100 in PBS for 5 min. After three times washing in PBS, blocking was performed by incubation in 5% normal goat serum in PBTX (PBS with 0.1% Triton X-100) at 37°C for 30 min. Incubation with primary antibodies was performed in 5% goat serum in PBTX at 37°C for 2 hr. Next, cells were washed three times with PBS and incubated for 1 hr at 37°C with secondary antibodies in 5% goat serum in PBTX. Then, after three times washing with PBS, cells were stained with 4′,6-diamidino-2-phenylindole (DAPI, 1:300, Invitrogen) at 37°C for 5 min in PBTX solution. The cells were washed three times with PBS, and the coverslips were mounted on glass slides with VECTASHIELD Antifade Mounting Medium (Vector Laboratories) and kept at 4°C before imaging.

## Immunocytochemistry of fly brain

The larval brains were dissected in PBS (pH 7.4), and fixed in PBS with 4% formaldehyde for 15 min. After the brains were washed in PBS containing 0.4% (v/v) Triton X-100 (PBTX) for 15 min three times, the brains were incubated with primary antibodies diluted in 0.4% PBTX with 5% normal goat serum overnight. Then, secondary antibodies were at room temperature for 1 hr, followed by DAPI (1:300, Invitrogen) staining for 10 min. Brains were mounted on glass slides with VECTASHIELD Antifade Mounting Medium (Vector Laboratories) and kept at 4°C before imaging.

## Confocal image acquisition and image analysis

Confocal microscopy was performed with an Olympus IX81 confocal microscope coupled with ×10, ×20 air lens or ×40, ×60 oil immersion objectives, and images were processed using FluoView 10-ASW (Olympus). Specifically, *Figure 7B and C* were analyzed using the ImageJ interactive 3D surface Plot plugin.

## Western blot analysis

Proteins were extracted from cells in RIPA (radioimmunoprecipitation assay) buffer 1 mM protease inhibitor cocktail (Sigma-Aldrich). Samples were heated at 100°C for 10 min in a 4× loading buffer. Proteins were separated on a Bis-Tris gel and transferred to nitrocellulose membranes. Then, membranes were blocked with blocking buffer (Rockland) for 1 hr at room temperature. Primary antibodies were incubated at 4°C overnight and secondary antibodies were incubated for 1 hr at room temperature. Images were processed on an Odyssey Infrared Imaging System or Amersham Imager 600 and analyzed using Image Studio software or ImageJ.

## Dot blot analysis

Proteins were extracted from cells in RIPA buffer 1 mM protease inhibitor cocktail (Sigma-Aldrich). Proteins were loaded with same amount on PVDF membranes. Then, membranes were blocked with Casine buffer for 1 hr at room temperature. Primary antibodies were incubated at 4°C overnight and secondary antibodies were incubated for 1 hr at room temperature. Images were processed on an Amersham Imager 600 and analyzed using ImageJ software.

## Cell proliferation test

Cells were seeded into the E-Plate 96 (ACEA) with the same confluence per well. Then, the plate was incubated at 37°C in 5% $CO_2$ for about 100 hr. The instrument was used to monitor the cell growth index. The cell growth curve was drawn with the value of each group from xCELLigence RTCA SP instrument.

## Colony formation assay

Cells were seeded with 1000 per well in a six-well plate containing 2 ml medium and replaced medium every 2 days. Cells were washed with 1 ml PBS three times and fixed with 1 ml formaldehyde for 15 min. After washed with PBS, cells were stained in 0.1% crystal violet buffer (Sigma-Aldrich) for 15 min. Cells were washed with pure water gently, and plates were put at room temperature to dry. Images were processed on an Amersham Imager 600 and analyzed using ImageJ software.

## Immunoprecipitation

Proteins were extracted from cells in RIPA buffer. Proteins were incubated with Protein-A beads (Thermo Fisher Scientific) conjugated with anti-p53 antibody or Mouse IgG at 4°C overnight with gentle shaking. After removing the supernatant, the bead pellets were collected and suspended with lysis buffer. Proteins were heated with loading buffer for 10 min at 100°C for loading to gel.

## Flow cytometry

Cells were prepared according to cell cycle and cell apoptosis detection kits (BD Pharmingen) after knockdown of NMNAT 72 hr. AnnexinV:PI gating was selected and analyzed to divide the data into quadrants, where Q3 was considered as viable, Q4 as early apoptosis, and Q2 as end stage apoptosis and death.

## Statistics

For each statistical test, biological sample size (n), and p-value are indicated in the corresponding figure legends. All data in this manuscript are shown as mean ± SD or median ± quartiles (specified in figure legends). t-test was used to compare between two groups, and one-way ANOVA with Bonferroni's post hoc test was applied to compare among three or more groups. Data were analyzed with Prism (GraphPad Software). Specifically, fly survival data were analyzed by the Chi-square test in R.

# Acknowledgements

The authors thank V Chavez Perez, Qin Yang, and Ling Zhang for technical expertise; and Joun Park for manuscript comments. The authors thank the FACS core facility at Sylvester Comprehensive Cancer Center, and Shannon Jacqueline Saigh for technical support. This research was supported by the Taishan Scholar Project of Shandong Province, the Science and Technology Support Program for Youth Innovation in Universities of Shandong (2019KJM009), the National Natural Science Foundation of China (82073888), the Top Talents Program for One Case Discussion of Shandong Province, Bohai rim Advanced Research Institute for Drug Discovety (LX211011), and the Sylvester Comprehensive Cancer Center.

# Additional information

## Funding

| Funder | Grant reference number | Author |
| --- | --- | --- |
| National Natural Science Foundation of China | 82073888 | Hongbo Wang |
| Science and technology support program for youth innovation in universities of Shandong | 2019KJM009 | Hongbo Wang |

| Funder | Grant reference number | Author |
|---|---|---|
| Sylvester Comprehensive Cancer Center | | Rong Grace Zhai Priyamvada Rai |
| Bohai Rim Advanced Research Institute for Drug Discovery (LX211011) | | Hongbo Wang |

The funders had no role in study design, data collection and interpretation, or the decision to submit the work for publication.

## Author contributions

Jiaqi Liu, Yi Zhu, Data curation, Formal analysis, Investigation, Methodology, Writing – original draft, Writing – review and editing; Xianzun Tao, Data curation, Investigation, Methodology, Writing – review and editing; Chong Li, Methodology, Resources, Writing – review and editing; Kai Ruan, Data curation, Methodology, Resources, Writing – review and editing; Zoraida Diaz-Perez, Investigation, Methodology; Priyamvada Rai, Data curation, Writing – review and editing; Hongbo Wang, Conceptualization, Funding acquisition, Project administration, Resources, Writing – review and editing; R Grace Zhai, Conceptualization, Data curation, Formal analysis, Funding acquisition, Investigation, Project administration, Supervision, Writing – original draft, Writing – review and editing

## Author ORCIDs

Jiaqi Liu http://orcid.org/0000-0001-6526-8655
Xianzun Tao http://orcid.org/0000-0002-5877-2391
Yi Zhu http://orcid.org/0000-0002-1778-8880
Priyamvada Rai http://orcid.org/0000-0001-7822-7553
Hongbo Wang http://orcid.org/0000-0001-8530-2339
R Grace Zhai http://orcid.org/0000-0002-7599-1430

## Decision letter and Author response

Decision letter https://doi.org/10.7554/eLife.70046.sa1
Author response https://doi.org/10.7554/eLife.70046.sa2

# Additional files

## Supplementary files

- Transparent reporting form
- Source data 1. Original data file.

## Data availability

All data generated or analysed during this study are included in the manuscript and supporting files. Source data files have been provided.

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
