## [Editor Report]

The authors found that NMNAT binds to p53 and that p53 is post-translationally regulated to control apoptosis in glioma models. Work found that depletion of NMNAT1 and NMNAT2 inhibits and that overexpression of the enzymes promotes glioma growth. Furthermore, depletion of NMNAT_1/2_ increases apoptosis and overexpression of the enzymes inhibits apoptosis upon cisplatin treatment. This is an exciting mechanism that extends NAD biology, p53 regulation, and the field of glioma pathogenesis.

---

## [Decision Letter]

[Editors’ note: the authors submitted for reconsideration following the decision after peer review. What follows is the decision letter after the first round of review.]

Thank you for submitting your work entitled "NMNAT promotes glioma growth through enhancing poly(ADP-ribosyl)ation of p53 to inhibit apoptosis" for consideration by *eLife*. Your article has been reviewed by 3 peer reviewers, and the evaluation has been overseen by a Reviewing Editor and a Senior Editor. The following individual involved in review of your submission has agreed to reveal their identity: Yun Fan (Reviewer #2).

Our decision has been reached after consultation between the reviewers. Based on these discussions and the individual reviews below, we regret to inform you that your work in its current form will not be considered for publication in *eLife*. However, the reviewers did find the work interesting and raised a number of substantial issues that I have summarised below. If you are able to address the issues in full, we would be happy to consider a much revised version of your paper as a new submission.

Summary:

This paper by the Zhai lab shows that NMNAT is necessary and sufficient for glioma growth in an in vivo fly model and in an in vitro cell culture model. The most exciting aspect of this work is the mechanistic insight showing that NMNAT is responsible for the parylation and acetylation of the tumor suppressor p53. The paper concludes that this action of NMNAT inhibits p53/caspase-mediated apoptosis and thus promotes glioma growth. This is new and of interest, but at the moment also too preliminary. The reviewers recommend a number of essential experiments. The full reviews are also included below and we welcome any additional (non-essential) work you believe you can add as well as textual and figure changes in response to their questions:

Essential issues:

1) Increase the quality of the parylation and acetylation blots by 'cleaning up' the bots, quantifying them and showing specificity:

The most mechanistic and exciting finding in this manuscript is the demonstration that overexpression of NMNAT led to increased parylation and decreased acetylation of p53. Both of these manipulations would inhibit p53-induced apoptosis. This is the most important result in the paper, but unfortunately the data are not convincing. There is no quantification of this effect. Worse yet, a single lane of each is shown, and is not even obvious by eye. First, the band that is claimed to be PAR-p53 is faint and is near the huge IgG band (get some beads with antibody covalently attached so you don't have this band). Second, this blot is full of bands-what is the proof that this band is PAR-p53? Finally, once the band is validated, then it must be repeated multiple times and quantified. The acetyl-p53 blot is also unconvincing. There are many bands, with no validation that the band labeled as acetyl-p53 is actually acetyl-p53. Again, a single blot is shown, and nothing is quantified. Indeed, even by eye it is not clear that NMNAT1 has any effect.

2) The biochemistry should also be done under conditions of NMNAT loss of function (in Figure 7):

There is a second issue with figure 7. Unlike the other studies in the manuscript, these experiments are only done with overexpression. The authors should test knockdown of the NMNATs in these paradigms. Since they have demonstrated that knockdown and overexpression of NMNATs have opposite effects on glioma cell growth, then if this is the mechanism, they should also have opposite effects on p53 posttranslational modifications. Of course, all of these experiments will require quantification.

3) To provide more direct evidence for the role of p53 and caspases in glioma development in *Drosophila*; both issues can be tested using *Drosophila* genetics:

a) Does apoptosis play a critical role to restrict RasV12-induced glioma development in *Drosophila*? Notably, the cleaved Caspase-3 antibody indicates the activity of Dronc, the initiator caspase in *Drosophila*, which has non-apoptotic functions. Also, a recent paper actually suggests that expression of RasV12 blocks activation of apoptosis in epithelial tissues (Perez et al., *eLife* 2017, 6: e26747). It is therefore important to show whether blocking apoptosis or caspase activity enhances glioma progression.

b) Test if is p53 required for activation of caspases in RasV12-induced glioma and in response to loss-of-NMNAT (Figure 4B)?

4) Properly discuss the limitations of the fly model (as indicated by the third reviewer), both the use of a Ras model in flies that is a model of only 2% of glioma and second to use the term glial neoplasia instead of glioma (or use glioma model).

5) Look in to TCGA or other databases (Human Protein Atlas, Ivy Foundation Atlas, Gliovis, etc) to check if NMNAT is upregulated in human tumors.

*Reviewer #1:*

This study investigates the role of the NAD biosynthetic enzyme NMNAT in glioma growth. While prior studies have highlighted the important role of the NAD biosynthetic pathway in glioma, this is the first to investigate NMNAT. Using both an in vivo fly model and in vitro cell culture model, the authors find that knockdown of NMNAT inhibits and overexpression of NMNAT promotes glioma growth. Since a role for NAD synthesis is expected, this is not a surprising finding. Indeed, I raise a number of issues below about the specificity of the findings. However, the authors go on to do a potentially interesting mechanistic analysis, claiming that NMNAT overexpression regulates the parylation and acetylation of p53. If these findings are validated (see #4 and #5 below), I think this will be an important and exciting contribution to the field.

1) In figure 2A the authors claim that the "increase in NMNAT" is required for glioma growth, but their experiment is to knockdown NMNAT. They don't make clear if this is back to wild type levels, or well below wild type levels. If well below, then they cannot conclude that it is the "increase" of NMNAT that matters-instead it just might be that you need NMNAT for a glioma to grow. Testing if a NMNAT heterozygote suppresses could test their hypotheses better, or carefully quantifying the effect of knockdown compared to wild types could also help.

2) More broadly, it is not surprising that you need NMNAT for glioma growth as you likely need many essential metabolic enzymes for fast growing cells to grow. It would be helpful to test the specificity of these findings. This should be done in two ways. First, is NMNAT necessary for healthy cells to grow. This could be checked by using the same RNAi in clones in imaginal discs-is growth inhibited? The second question to address is whether there is anything selective about NMNAT loss. Do other genes in the NAD biosynthetic pathway also block glioma growth?

3) Figure 3 addresses the same issue of NMNAT dependence in a human glioma cell line model. Specificity issues again apply. Why does knockdown of either of two NMNAT genes give the same strong phenotype? The authors make a big point about compartmentalization of NAD biosynthesis, and yet here they get the same phenotype from the nuclear and cytoplasmic variants of NMNAT. What are the effects of each of these manipulations on the level of NAD? Does an equivalent drop in NAD induced by inhibiting the pathway via other means (using Nampt inhibitors) give the same phenotype?

4) The most mechanistic and exciting finding in this manuscript is the demonstration that overexpression of NMNAT led to increased parylation and decreased acetylation of p53. Both of these manipulations would inhibit p53-induced apoptosis. This is the most important result in the paper, but unfortunately the data are not convincing. There is no quantification of this effect. Worse yet, a single lane of each is shown, and is not even obvious by eye. First, the band that is claimed to be PAR-p53 is faint and is near the huge IgG band (get some beads with antibody covalently attached so you don't have this band). Second, this blot is full of bands-what is the proof that this band is PAR-p53? Finally, once the band is validated, then it must be repeated multiple times and quantified. The acetyl-p53 blot is also unconvincing. There are many bands, with no validation that the band labeled as acetyl-p53 is actually acetyl-p53. Again, a single blot is shown, and nothing is quantified. Indeed, even by eye it is not clear that NMNAT1 has any effect.

5) There is a second issue with figure 7. Unlike the other studies in the manuscript, these experiments are only done with overexpression. The authors should test knockdown of the NMNATs in these paradigms. Since they have demonstrated that knockdown and overexpression of NMNATs have opposite effects on glioma cell growth, then if this is the mechanism, they should also have opposite effects on p53 posttranslational modifications. Of course, all of these experiments will require quantification.

*Reviewer #2:*

This manuscript addressed an important knowledge gap in the field – roles of the Nicotinamide mononucleotide adenylyltransferase (NMNAT) in promoting glioma progression, by using both *Drosophila* in vivo models and cultured human glioma cells.

The authors first determined that NMNAT is upregulated in glioma induced by activated Ras (RasV12) in *Drosophila*. Further loss- and gain-of-function analyses showed that NMNAT, particularly its nuclear isoform, inhibits nuclear expression of Caspases, the proteases activating apoptosis, and p53, the key mediator of DNA damage responses, therefore promotes glioma development. From these, the authors conclude that NMNAT accelerates glioma progression by inhibiting p53/caspase-mediated apoptosis. Importantly, although not exactly same in my view, similar principles underlying functions of NMNAT apply to human glioma cells. The authors convincingly showed that, in these cells, NMNAT suppresses the nuclear enrichment of Caspase-3 and activation of apoptosis. Interestingly, NMNAT regulates activity of p53 via posttranslational modifications including PARylation and acetylation, but not the total level of p53. Overall, these findings are novel and are of interest to the broad readership of *eLife*.

However, direct evidence showing roles of p53 and caspases in glioma development in *Drosophila* is missing in the manuscript. To my knowledge, these have not been characterized in the field hence are critical to support the conclusions made.

Along this line, my concerns are as follows:

1) Does apoptosis play a critical role to restrict RasV12-induced glioma development in *Drosophila*? Notably, the cleaved Caspase-3 antibody indicates the activity of Dronc, the initiator caspase in *Drosophila*, which has non-apoptotic functions. Also, a recent paper actually suggests that expression of RasV12 blocks activation of apoptosis in epithelial tissues (Perez et al., *eLife* 2017, 6: e26747). It is therefore important to show whether blocking apoptosis or caspase activity enhances glioma progression.

2) Is p53 required for activation of caspases in RasV12-induced glioma and in response to loss-of-NMNAT (Figure 4B)?

*Reviewer #3:*

NMNAT promotes glioma growth through enhancing poly(ADP-ribosyl)ation of p53 to inhibit apoptosis – Paper Review Using a *Drosophila* glioma model and follow-up studies in human glioma cell lines, Liu et al. uncovered the requirement of NMNAT in glioma progression.

One concern is the genetics of the model system chosen. This manuscript focuses on Ras using a *Drosophila* model. However, mutations in Ras proteins, such as HRAS, are limited in human glioma tumors. In omic studies such as TCGA, there are very few genomic alterations in human glioma (2% or less in HRAS, KRAS, and NRAS). This is likely because driver alterations affecting the RAS pathway lie upstream of RAS itself, and this is different than other solid tumors, as the authors cite in the manuscript. This significantly limits the scope and impact of the findings and should be addressed in some way.

Another concern is the authors use of glioma nomenclature. *Drosophila* models do not get true "gliomas" as defined by neuropathological criteria for humans and other animals, so the language in the paper describing their glioma model should change to "neoplastic glia," "glial neoplasia" or add the qualifier "model" in their sentences to better distinguish between human and fly biology for this disease. Flies can only model certain aspects of gliomas, such as the genetic basis of neoplastic glial transformation.

Is NMNAT upregulated in tumor specimens from human glioma patients? The authors could query TCGA or other databases (Human Protein Atlas, Ivy Foundation Atlas, Gliovis, etc) to compare the expression levels of NMNAT mRNA and protein between gliomas of different grade and between gliomas with and without mutations in Ras pathway components. These databases are publicly available and this analysis could be done without any benchwork. Or, if they are able to do benchwork, they could compare protein levels in normal human neural stem cells or astrocytes vs glioma cell lines.

This study may benefit from additional studies on DNA damage such as a comet assay or even looking at whether there are cell cycle changes as a result of modified p53 function.

Figure 2E – Manuscript states that Nmnat-PC increased lethality, but the graph is for survival at different life stages. This is confusing, particularly because some of the samples exceed 100% with the way the error bars are presented. This graph should be reformatted to show % lethal broken down by lethal stage.

Figure 3 – Colony formation assays are not sufficient to demonstrate growth defects in the glioma cells. Additional independent methods that support these results should be shown. FACS profiling would be appropriate. Also, at least one other glioma cell line that is Ras-dependent should be tested and results from those cells should be shown as well. Why were T98G cells chosen? What is the rationale for using those cells? Were there any other cells tested that yielded negative results? If so, results from those cells may be informative if they are P53 null (many glioma cell lines are P53 null).

Figure 5B – this is an unusual way to quantify cell death in glioma cells. A more representative way would be to quantify the percentage of caspase-positive cells rather than the intensity of caspase staining per cell, which is a much more transient and variable thing. Perhaps the authors already have this data and can change the parameters of their data analysis?

Figure 7A – I would argue only NMNAT2 overexpression is the only condition that increases p53 PARylation. Band intensity could be quantified to better support the author's conclusions.

Figure S1: There is no criteria set forth to define pre-tumor vs. glioma areas. Are there molecular markers for these regions? Are those markers conserved between human and flies? The presence of excess cells in some regions of the brain is not really sufficient to distinguish between pre-neoplastic and neoplastic cells in the fly brain.

[Editors’ note: the authors submitted for reconsideration following the decision after peer review. What follows is the decision letter after the second round of review.]

Thank you for resubmitting the paper entitled "NMNAT promotes glioma growth through regulating post-translational modifications of p53 to inhibit apoptosis" for further consideration by *eLife*. Your revised article has been evaluated by a Senior Editor and a Reviewing Editor. We are sorry to say that we have decided that this submission will not be considered further for publication by *eLife*.

The revised paper has been seen by a reviewer who can judge the *Drosophila* work in your manuscript and by a reviewer who can judge the p53 biology. In the discussion among the reviewers there were serious concerns regarding tools and conclusions that pertain to the p53 biology that you present to the point that the reviewers recommended to not move forward with the manuscript. More specifically, the last section of reviewer 2's main comments that the p53 in glioma cell lines that you used differ in their p53 status (based on published literature, we couldn't find that you checked this yourself), but show a similar phenotype, and that this is not compatible with the model you suggest was a the reason for rejection. I realise this is a revised version and I am very sorry to have to bring this news to you.

*Reviewer #1:*

The authors have worked very hard to address all of my concerns and those of the other reviewers. The manuscript is much improved and much more convincing.

I am satisfied with the changes made in this revised version.

*Reviewer #2:*

Liu et al. investigate the role of the NAD+ synthases NMNAT1 and NMNAT2 in glioma survival to test whether these enzymes are promising targets for therapeutic interventions. The authors employ a glial neoplasia developing *Drosophila* model and the two human glioma cell line models T98G and U87MG. Their data show that depletion of NMNAT1 and NMNAT2 inhibits and overexpression of the enzymes promotes glioma growth. Moreover, the authors show that depletion of NMNAT_1/2_ increased apoptosis while overexpression of the enzymes inhibited apoptosis upon cisplatin treatment. Mechanistically, the authors propose that NMNAT_1/2_ physically interact with p53 and PARP1 to drive PARylation and deacetylation of p53, ultimately leading to reduced p53 activity.

I find substantial parts of the underlying data not convincing (see below). I felt that the authors overinterpreted poor or insufficiently controlled data, jumped to conclusions from ambiguous results, and exaggerated several conclusions.

Together, I cannot recommend the manuscript for publication.

I am not fully convinced by the growth data presented by the authors. Publicly available data (www.DepMap.org) do not indicate that glioma cell lines are particularly sensitive to NMNAT1 or NMNAT2 depletion. Controls for si-mediated protein reduction in the cell lines are missing throughout and a second siControl could be helpful.

The PARylation data is not convincing. A methods part is missing that explains how PARylation was analyzed. Total PAR or PAR overlay assay? The latter would be crucial to claim p53 PARylation, the former is critical for statements on PARylation in general. The authors claim that NMNAT_1/2_ form a complex with p53 and PARP1 to facilitate PARylation, but it is unclear whether they refer to only p53 PARylation or also PARylation in general.

The authors draw conclusions on p53 activity from signals that are known to be regulated by various pathways (e.g. apoptosis). Instead, it would be crucial to test for the expression of p53 targets (e.g. MDM2, BBC3, GADD45A) to investigate p53 activity. Yet, given that the p53 status differs between T98G (mutant p53) and U87MG (wild-type p53) cells (van Meir et al., 1994 Cancer Res), the authors' growth data, which are similar for T98G and U87MG, demonstrate that the potential effect elicited by NMNAT_1/2_ is independent of. Moreover, all p53 interaction data shown using T98G data reflect mutant p53, which is known to have potentially different binding partners than wild-type p53. In general, the fact that T98G harbor mutant p53 turns most parts of the manuscript upside down that deal with mechanistics.

Figure 1: why is Nmnat not color-coded?

Figure 3G: The AnnexinV:PI gating is strange. For comparison see https://images.bio-rad-antibodies.com/kit/annexin-v-kit-antibody-kit-annex100-image2-600w.jpg.

Western blot signals are often saturated, which hinders proper readout.

Figures 5A, B, D, and F miss NMNAT1 and NMNAT2 as crucial controls.

Figures 5D and F: All lanes display cisplatin-treated samples? If so, controls without cisplatin treatment are missing.

Figure 8A and Figure 8—figure supplement 1: PARylation of what? Controls are missing.

Figure 8C: Specificity of the NMNAT2 antibody is not convincing.

Figure 9A: It is unclear why the authors IP p53 to blot acetyl-p53. Acetyl-p53 can be immunoblotted with whole cell lysates, as demonstrated by the authors (input lanes). Loading of IPs is more difficult to control, thus the signal from the input lanes are more important and actually do not show changes in acetyl-p53 upon NMNAT overexpression.

Figure 9C: What is quantified here? 9B apparently is based on the p53 IP lanes. But 9C appears to be based on the input lanes. Cherry picking?

Figure 3—figure supplement 3: NMNAT2 blot is out of focus.

Figure 5—figure supplement 2: reduction of cleaved cas3 upon NMNAT1 overexpression is not convincing. siNMNAT_1/2_ are missing. Immunoblots for NMNAT_1/2_ are missing.

Figure 9—figure supplement 1: What is shown in this Figure? Whole cell lysates or p53 IPs? Why do the authors use only lysate or IP here but both in Figure 9A? The reduction is not convincing. No cisplatin control is missing. siNMNAT_1/2_ are missing.

Immunoblots for NMNAT_1/2_ are missing.

[Editors’ note: further revisions were suggested prior to acceptance, as described below.]

Thank you for resubmitting your work entitled "NMNAT promotes glioma growth through regulating post-translational modifications of p53 to inhibit apoptosis" for further consideration by *eLife*. Your revised article has been evaluated by Jonathan Cooper (Senior Editor) and a Reviewing Editor.

There is consensus that the manuscript has been much improved but there are some remaining issues that need to be addressed, as outlined below:

We would like you to submit a final paper that includes:

1) Data indicating that NMNAT_1/2_ interacts with p53 also in the U87MG p53 wild-type line;

2) Data that tests if p53 depletion rescues the apoptosis-inducing effect of siNMNAT in both cell lines they use (T98G and U87MG);

3) Toning down claims on PARylation at different locations in the manuscript.

*Reviewer #2:*

The authors provided a revised version addressing most concerns that I raised. However, some key points in the authors' study still require supportive data.

– The NMNAT-PARP1-p53 interaction:

Previously, I raised the concern that protein interaction with mutant p53 not always translates into an interaction with wild-type p53. The present M237I mutant reportedly possesses neomorphic functions. Given that the NMNAT_1/2_-p53 interaction is an integral part of the model proposed by the authors, I would like to reiterate that I find it crucial to corroborate this interaction also in the p53 wild-type line U87MG.

– The NMNAT-p53-apoptosis mechanism:

Following my concern that much of the authors' data on p53 was generated using a mutant p53 cell line (T98G), the authors explain that they intentionally included both wild-type and mutant p53 cell systems and clarify their strategy to the reader in the revised version, such as by adding a respective sentence to the Discussion section "Our findings that NMNAT similarly affects p53 modification in either wild-type (U87MG) or mutant p53 (T98G) cells suggest NAD+-dependent PARylation or deacetylation of p53 is independent of the p53 [M237I] mutation. Indeed recent studies have shown that mutant p53 proteins retain the ability to induce apoptosis despite losing tumor-suppressive transactivation functionality (Timofeev et al., 2019)".

Notably, Timofeev et al., 2019 refers to the p53 mutant R181E (R178E in mice). p53 mutants are known to differ. The M237I mutant present in the authors' T98G cell line, for example, did not display residual apoptosis-driving function when it was tested in a different study (Boettcher et al., 2019 Science). Given that the authors propose reduced p53-dependent apoptosis to be a key mechanism by which NMNAT promotes glioma growth, it is crucial to show that there actually is p53-dependent apoptosis occurring in the authors' experimental setup, i.e. by adding data on sip53 in Figure 3G showing whether p53 depletion can indeed rescue the apoptosis-inducing effect of siNMNAT in both T98G and U87MG cells. Given the importance to the authors' mechanistic model and the different experimental setup, it is insufficient to only refer to the findings by Enns et al., 2004.

– The NMNAT-p53-PARylation mechanism:

The authors convincingly demonstrate that NMNAT_1/2_ affect total PAR levels in the cell (Figure 8A and B, Figure 8-supplement 1). Key points in the authors' model include (1) that PARylation of p53 is induced by NMNAT_1/2_ (abstract, headlines, Figure 10E) and (2) that complex formation with p53 is important for NMNAT_1/2_ to facilitate PARylation (abstract). Supportive data for these points, however, is missing.

1) I would like to reiterate that it is crucial to provide data on NMNAT-dependent p53 PARylation. I.e. by blotting for PAR in p53 IPs (Figure 8C or 9A), in both T98G and U87MG.

2) To support the authors' point that "NMNAT forms a complex with p53 and PTM enzyme PARP1 to facilitate PARylation" (abstract), it is crucial to show whether p53 indeed is required for NMNAT-dependent PARylation, i.e. whether p53 depletion affects NMNAT-dependent PARylation in both T98G and U87MG (Figures 8A and 8-supplement 1).

---

## [Author Response]

[Editors’ note: the authors resubmitted a revised version of the paper for consideration. What follows is the authors’ response to the first round of review.]

Essential issues:1) Increase the quality of the parylation and acetylation blots by 'cleaning up' the bots, quantifying them and showing specificity:The most mechanistic and exciting finding in this manuscript is the demonstration that overexpression of NMNAT led to increased parylation and decreased acetylation of p53. Both of these manipulations would inhibit p53-induced apoptosis. This is the most important result in the paper, but unfortunately the data are not convincing. There is no quantification of this effect. Worse yet, a single lane of each is shown, and is not even obvious by eye. First, the band that is claimed to be PAR-p53 is faint and is near the huge IgG band (get some beads with antibody covalently attached so you don't have this band). Second, this blot is full of bands-what is the proof that this band is PAR-p53? Finally, once the band is validated, then it must be repeated multiple times and quantified. The acetyl-p53 blot is also unconvincing. There are many bands, with no validation that the band labeled as acetyl-p53 is actually acetyl-p53. Again, a single blot is shown, and nothing is quantified. Indeed, even by eye it is not clear that NMNAT1 has any effect.

Measured PARylation level and quantified. (Fig. 8 A, B). Showed specific acetylated p53 blots and quantified. (Fig. 9 and Fig. 9_figure supplement 1).

2) The biochemistry should also be done under conditions of NMNAT loss of function (in Figure 7):There is a second issue with figure 7. Unlike the other studies in the manuscript, these experiments are only done with overexpression. The authors should test knockdown of the NMNATs in these paradigms. Since they have demonstrated that knockdown and overexpression of NMNATs have opposite effects on glioma cell growth, then if this is the mechanism, they should also have opposite effects on p53 posttranslational modifications. Of course, all of these experiments will require quantification.

Measured PARylation after Knockdown of NMNAT1/2. (Fig. 8_figure supplement 1).

3) To provide more direct evidence for the role of p53 and caspases in glioma development in *Drosophila*; both issues can be tested using *Drosophila* genetics:a) Does apoptosis play a critical role to restrict RasV12-induced glioma development in *Drosophila*? Notably, the cleaved Caspase-3 antibody indicates the activity of Dronc, the initiator caspase in *Drosophila*, which has non-apoptotic functions. Also, a recent paper actually suggests that expression of RasV12 blocks activation of apoptosis in epithelial tissues (Perez et al., eLife 2017, 6: e26747). It is therefore important to show whether blocking apoptosis or caspase activity enhances glioma progression.

Examined the functional effects of Caspase on *Ras^v12^* induced glioma. (Fig. 4_figure supplement 1). Inhibited caspase pathway genetically in *Ras^v12^* overexpressing fly and determined the consequence by glial neoplasia volume.

b) Test if is p53 required for activation of caspases in RasV12-induced glioma and in response to loss-of-NMNAT (Figure 4B)?

Examined the effects of p53 on *Ras^v12^* induced glioma. (Fig. 7) p53 inhibitor feeding in *Ras^v12^* overexpressing fly, and determined the consequence by glial neoplasia volume, survival and cleaved caspase activity.

4) Properly discuss the limitations of the fly model (as indicated by the third reviewer), both the use of a Ras model in flies that is a model of only 2% of glioma and second to use the term glial neoplasia instead of glioma (or use glioma model).

1. Added in the discussion part.

2. Used the correct nomenclature of “glial neoplasia” instead of “glioma” in the text.

5) Look in to TCGA or other databases (Human Protein Atlas, Ivy Foundation Atlas, Gliovis, etc) to check if NMNAT is upregulated in human tumors.

Added figure supplement. (Fig. 10_figure supplement 1-2)

Reviewer #1:This study investigates the role of the NAD biosynthetic enzyme NMNAT in glioma growth. While prior studies have highlighted the important role of the NAD biosynthetic pathway in glioma, this is the first to investigate NMNAT. Using both an in vivo fly model and in vitro cell culture model, the authors find that knockdown of NMNAT inhibits and overexpression of NMNAT promotes glioma growth. Since a role for NAD synthesis is expected, this is not a surprising finding. Indeed, I raise a number of issues below about the specificity of the findings. However, the authors go on to do a potentially interesting mechanistic analysis, claiming that NMNAT overexpression regulates the parylation and acetylation of p53. If these findings are validated (see #4 and #5 below), I think this will be an important and exciting contribution to the field.

We thank the reviewer for recognizing the significance of our findings. In this revised version, we have comprehensively characterized the cellular and biochemical mechanisms underlying the effects of NMNAT on promoting glioma growth in both fly model and human glioma cell model. Specifically, we identified the role of NMNAT in regulating the NAD^+^-dependent post-translational modifications of p53: PARylation and deacetylation.

1) In figure 2A the authors claim that the "increase in NMNAT" is required for glioma growth, but their experiment is to knockdown NMNAT. They don't make clear if this is back to wild type levels, or well below wild type levels. If well below, then they cannot conclude that it is the "increase" of NMNAT that matters-instead it just might be that you need NMNAT for a glioma to grow. Testing if a NMNAT heterozygote suppresses could test their hypotheses better, or carefully quantifying the effect of knockdown compared to wild types could also help.

We would like to clarify that our data support the conclusion that “NMNAT is required for glioma growth”. This is supported by both loss of function (knockdown) and gain of function (overexpression) experiments.

In the knockdown experiment, we carefully analyzed the knockdown level following the reviewer’s suggestion and found that the level of NMNAT in flies with *Ras^v12^* overexpression and NMNAT RNAi is lower than that of wild-type flies (Figure 2_figure supplement 3).

In the revised manuscript, we clarified the conclusion to avoid confusion. The new version reads; on page 7, “These results suggest that nuclear enzymatically active NMNAT promoted glial neoplasia growth.”

2) More broadly, it is not surprising that you need NMNAT for glioma growth as you likely need many essential metabolic enzymes for fast growing cells to grow. It would be helpful to test the specificity of these findings. This should be done in two ways. First, is NMNAT necessary for healthy cells to grow. This could be checked by using the same RNAi in clones in imaginal discs-is growth inhibited? The second question to address is whether there is anything selective about NMNAT loss. Do other genes in the NAD biosynthetic pathway also block glioma growth?

We thank the reviewer for bringing up this important point. Indeed, it is expected that metabolic enzymes are essential for cell growth, especially fast-growing cancer cells. However, identifying the molecular mechanism of the requirement is critical for targeted anti-cancer therapy. Our previous work has shown that NMNAT as NAD^+^ biosynthetic enzyme is essential for organism survival, as loss of NMNAT causes organismal lethality but not cell death (Zhai et al., 2006). We carried out the RNAi experiment as suggested by the reviewer and found that RNAi-mediated knockdown of NMNAT in normal glial cells (without Ras^v12^ expression) did not result in growth inhibition (Figure 2_figure supplement 1)**,** suggesting NMNAT is not essential for healthy cell survival.

Regarding the second question on other genes in the NAD^+^ biosynthetic pathway for glioma growth, significant research effort has been invested in targeting the NAD^+^ synthetic pathway, especially the rate-limiting enzyme NAMPT. Several studies have shown that NAMPT is critical for glioma progression (Guo et al., 2019; Lucena-Cacace, Otero-Albiol, Jimenez-Garcia, Peinado-Serrano, and Carnero, 2017; Lucena-Cacace, Umeda, Navas, and Carnero, 2019). However, the clinical outcomes of NAMPT inhibitors as chemotherapy have been largely disappointing, which begs the mechanist investigation of the NAD^+^ pathway in glioma and the consideration of alternative targets (Sampath, Zabka, Misner, O'Brien, and Dragovich, 2015). We included this information in the introduction, as this very point is the primary motivation of our work.

In the revised manuscript, we extended the description of this point in the introduction, on page 4, in the paragraph starts with “Disappointingly, several clinical trials of NAMPT inhibitors have failed due to low efficacy and high toxicities, which demands the urgent consideration of an alternative target in the NAD^+^ metabolic pathway.”

3) Figure 3 addresses the same issue of NMNAT dependence in a human glioma cell line model. Specificity issues again apply. Why does knockdown of either of two NMNAT genes give the same strong phenotype? The authors make a big point about compartmentalization of NAD biosynthesis, and yet here they get the same phenotype from the nuclear and cytoplasmic variants of NMNAT. What are the effects of each of these manipulations on the level of NAD? Does an equivalent drop in NAD induced by inhibiting the pathway via other means (using Nampt inhibitors) give the same phenotype?

The reviewer touched on an important point. The concept of local/compartmentalized NAD^+^ synthesis has been proposed based on two main observations, first, the relatively short half-life and diffusion radius of NAD^+^, and second, the compartment-specific localization of several NAD^+^ synthetic enzymes including NMNAT. Precise measurement of the compartmental concentration of NAD^+^ has been challenging (Rechsteiner, Hillyard, and Olivera, 1976; van Roermund, Elgersma, Singh, Wanders, and Tabak, 1995). It is estimated that mitochondria have the highest level of NAD^+^, around 250 uM, while the nucleus and cytosol share a similar concentration of NAD^+^, around 100 uM, given the large size of nuclear pores that is permissible for the exchange of NAD^+^ and metabolic substrates (Alano et al., 2007; Camacho-Pereira et al., 2016). Furthermore, it has been shown that in mammalian cells, nuclear and cytoplasmic NMNATs can regulate each other’s activity, likely through feedback from a dynamic pool of substrates NMN and ATP, as overexpressing cytoplasmic NMNAT may exhaust the supply of NMN, therefore, repress nuclear NAD^+^ synthesis (Ryu et al., 2018). These studies highlight the complexity of the regulation of cellular NAD^+^ levels, which includes the contribution from localized synthetic and catabolic enzymes, as well as the dynamic influences from neighboring cellular compartments.

To probe at this complexity, we included both nuclear and cytoplasmic NMNATs in all of our analyses in both *Drosophila* and glioma cell lines experimental models. We found that in the *Drosophila* model in vivo, nuclear NMNAT (PC) showed a visibly stronger effect in promoting cell growth than cytoplasmic NMNAT (PD), while knocking down either of two NMNAT genes in glioma cell lines resulted in similar effects of inhibiting cell growth. It is possible that the different effects between nuclear and cytoplasmic NMNATs were below the detection limit of our analysis. Taken together, our results suggest that localized NAD^+^ biosynthesis is important to glioma tumorigenesis, and there is a dynamic and complex interaction between the cytoplasmic and nuclear pools of NAD^+^ metabolites.

As mentioned in the comment addressing the previous concern (#2), NAMPT inhibitors have been explored as chemotherapy agents but did not result in clinical success (Sampath et al., 2015). While NAMPT is a uni-directional enzyme, synthesizing NMN (nicotinamide mononucleotide), NMNAT is downstream of NAMPT and directly regulates the level of NAD^+^ by catalyzing the reversible reaction of NAD^+^ synthesis. The direction of the reaction, forward (NAD^+^ production) or reverse (NAD^+^ breakdown), is dependent upon the availability of subtracts. Therefore, NMNAT functions as a cellular metabolic sensor and maintains the homeostasis of NAD^+^ pools. NAMPT inhibition would only result in a reduction in NMN and subsequently NAD^+^, NMNAT inhibition could have much more complex consequences of metabolic homeostasis. For these reasons, we did not include NAMPT inhibitors in our study but rather focused on NMNAT and its downstream functional consequences.

In the revised manuscript, we extended the discussion in addressing this series of questions in the Discussion section, on page 14, in the paragraph starts with “While NAMPT is a uni-directional enzyme, synthesizing NMN (nicotinamide mononucleotide), NMNAT is downstream of NAMPT and directly regulates the level of NAD^+^ by catalyzing the reversible reaction of NAD^+^ synthesis.”

4) The most mechanistic and exciting finding in this manuscript is the demonstration that overexpression of NMNAT led to increased parylation and decreased acetylation of p53. Both of these manipulations would inhibit p53-induced apoptosis. This is the most important result in the paper, but unfortunately the data are not convincing. There is no quantification of this effect. Worse yet, a single lane of each is shown, and is not even obvious by eye. First, the band that is claimed to be PAR-p53 is faint and is near the huge IgG band (get some beads with antibody covalently attached so you don't have this band). Second, this blot is full of bands-what is the proof that this band is PAR-p53? Finally, once the band is validated, then it must be repeated multiple times and quantified. The acetyl-p53 blot is also unconvincing. There are many bands, with no validation that the band labeled as acetyl-p53 is actually acetyl-p53. Again, a single blot is shown, and nothing is quantified. Indeed, even by eye it is not clear that NMNAT1 has any effect.

We thank the reviewer for the suggestion. Indeed, the post-translational modification of p53 is one of the most exciting findings, and also the most technically challenging. Due to their transient nature, the PTMs could be difficult to detect consistently. We took the reviewer’s criticism to heart and have expanded the biochemical characterization of NAD^+^-dependent PTMs, PARylation, and deacetylation. We employed multiple complementary approaches for each PTM and the new results were included in Figure 8, Figure 9, and Figure 9_figure supplement 1.

For PARylation, since there is no antibody available to specifically detect PAR-p53, we used two approaches to address this question. First, we assessed total PARylation level in cells with or without NMNAT overexpression and found that total PARylation is increased with NMNAT overexpression (Figure 8 A, B), suggesting that NMNAT promotes protein PARylation in general. Second, to probe the probability of p53 PARylation, we immunoprecipitated p53 and probed for the PARylation enzyme PARP1 (Figure 8C). We found that the p53, PARP1, and NMNAT forms a trimeric complex that was stable in immunoprecipitation. Importantly we found that endogenous PARP1 expression was upregulated in NMNAT overexpressing cells suggesting a co-regulation of NMNAT and PARP1 to facilitate the NAD^+^-dependent PARylation. These biochemical results were corroborated by the immunofluorescent imaging analysis where cells overexpressing NMNAT showed a higher level of PARP1 colocalizing with p53 (Figure 8 D-I). Together, these results support our hypothesis that p53 PARylation is increased in NMNAT overexpressing cells.

For NAD^+^-dependent deacetylation, we took advantage of an available acetyl-p53 specific antibody that detects acetylated p53 at residue K382. The NAD^+^-dependent deacetylation of p53 at K382 is mediated by SIRT1 (Cheng et al., 2003; Vaziri et al., 2001). We observed a significant reduction of acetylated-p53 levels in NMNAT overexpressing cells, and interestingly, a concomitant increase of SIRT1. We have added a new figure (Figure 9) to show the western analysis and quantifications. Our results suggest a co-regulation of NMNAT and SIRT1 to facilitate the NAD^+^dependent deacetylation of p53. To rule out a potential cell-line specific effect, we examined the acetylation of p53 in another human glioma cell line U87MG and observed a similar reduction in acetylp53 in NMNAT expressing cells (Figure 9_figure supplement **1**).

5) There is a second issue with figure 7. Unlike the other studies in the manuscript, these experiments are only done with overexpression. The authors should test knockdown of the NMNATs in these paradigms. Since they have demonstrated that knockdown and overexpression of NMNATs have opposite effects on glioma cell growth, then if this is the mechanism, they should also have opposite effects on p53 posttranslational modifications. Of course, all of these experiments will require quantification.

To address this point raised by the reviewer, we carried out NMNAT knockdown experiments and examine the cellular consequences of NMNAT reduction and found reduction of PARylation in siRNA knockdown cells. The results and quantification were included in a new figure (Figure 8_figure supplement 1).

Reviewer #2:This manuscript addressed an important knowledge gap in the field – roles of the Nicotinamide mononucleotide adenylyltransferase (NMNAT) in promoting glioma progression, by using both *Drosophila* in vivo models and cultured human glioma cells.The authors first determined that NMNAT is upregulated in glioma induced by activated Ras (RasV12) in *Drosophila*. Further loss- and gain-of-function analyses showed that NMNAT, particularly its nuclear isoform, inhibits nuclear expression of Caspases, the proteases activating apoptosis, and p53, the key mediator of DNA damage responses, therefore promotes glioma development. From these, the authors conclude that NMNAT accelerates glioma progression by inhibiting p53/caspase-mediated apoptosis. Importantly, although not exactly same in my view, similar principles underlying functions of NMNAT apply to human glioma cells. The authors convincingly showed that, in these cells, NMNAT suppresses the nuclear enrichment of Caspase-3 and activation of apoptosis. Interestingly, NMNAT regulates activity of p53 via posttranslational modifications including PARylation and acetylation, but not the total level of p53. Overall, these findings are novel and are of interest to the broad readership of eLife.However, direct evidence showing roles of p53 and caspases in glioma development in *Drosophila* is missing in the manuscript. To my knowledge, these have not been characterized in the field hence are critical to support the conclusions made.

We thank the reviewer for recognizing the significance of our findings. We have followed the reviewer’s recommendation and carried out additional analyses on the roles of p53 and caspase in the *Drosophila* model. The new results were included in the new figures (Figure 4_figure supplement 1 and Figure 7). These additions have greatly strengthened the mechanistic analysis and further supported our findings.

Along this line, my concerns are as follows:1) Does apoptosis play a critical role to restrict RasV12-induced glioma development in *Drosophila*? Notably, the cleaved Caspase-3 antibody indicates the activity of Dronc, the initiator caspase in *Drosophila*, which has non-apoptotic functions. Also, a recent paper actually suggests that expression of RasV12 blocks activation of apoptosis in epithelial tissues (Perez et al., eLife 2017, 6: e26747). It is therefore important to show whether blocking apoptosis or caspase activity enhances glioma progression.

We thank the reviewer for bringing up this important point. We have followed the suggestion and performed a series of genetic experiments in *Drosophila*. We blocked apoptosis or caspase activity in *Ras^v12^* overexpressing glial tissue by either overexpressing apoptosis inhibitor Diap1, or p35; or knocking down apoptosis initiator Dronc (Dronc-RNAi) and DCP1 (DCP1-RNAi). We next quantified glial neoplasia volume and found that indeed blocking apoptosis or caspase activity enhances glial neoplasia progression. We included the results in a new figure, Figure 4_figure supplement 1.

Regarding the issue with the caspase-3 antibody, we would like to point out that the caspase-3 antibody we used (Figure 4) recognizes both pro-caspase-3 (non-apoptotic activity) and cleaved caspase3 (apoptotic activity). As shown in Figure 4_figure supplement 1**,** the caspase pathway is conserved in mammals and fly. Dronc is homologue of caspase-9 in human. The caspase-3 antibody we used cannot recognize Dronc. We quantified the nuclear caspase-3 as the activated caspase-3 to indicate apoptotic levels.

2) Is p53 required for activation of caspases in RasV12-induced glioma and in response to loss-of-NMNAT (Figure 4B)?

We thank the reviewer for raising this interesting question. To answer this question, we carried out a p53 inhibitor feeding experiment in fly to assess the requirement of p53 in the activation of caspase in *Ras^v12^*-induced glial neoplasia. We found that inhibiting p53 promoted glial neoplasia volume, reduced cleaved caspase-3 levels, and reduced fly survival rate (Figure 7), suggesting the requirement of p53 in the activation of caspases in *Ras^V12^*-induced glial neoplasia.

We attempted to carry out the experiment of inhibiting p53 in *Ras^V12^*-induced glial neoplasia in the loss of NMNAT background, however, this combination of genetic manipulation resulted in early lethality that precluded any meaningful analysis.

Collectively, our results suggest that p53 is required for activation of caspases in *Ras^V12^*-induced glial neoplasia and p53 inhibition phenocopies NMNAT (nuclear PC) overexpression in promoting glial neoplasia tumorigenesis. We included this important result in a new figure in the revision (Figure 7).

Reviewer #3:NMNAT promotes glioma growth through enhancing poly(ADP-ribosyl)ation of p53 to inhibit apoptosis – Paper Review Using a *Drosophila* glioma model and follow-up studies in human glioma cell lines, Liu et al. uncovered the requirement of NMNAT in glioma progression.One concern is the genetics of the model system chosen. This manuscript focuses on Ras using a *Drosophila* model. However, mutations in Ras proteins, such as HRAS, are limited in human glioma tumors. In omic studies such as TCGA, there are very few genomic alterations in human glioma (2% or less in HRAS, KRAS, and NRAS). This is likely because driver alterations affecting the RAS pathway lie upstream of RAS itself, and this is different than other solid tumors, as the authors cite in the manuscript. This significantly limits the scope and impact of the findings and should be addressed in some way.

We thank the reviewer for pointing out the concern on the RAS driver. We agree that indeed RAS alterations in human glioma occur in much lower frequency than some high alteration genes such as IDH and EGFR. Our rationale for using mutant RAS overexpressing model in *Drosophila* was to probe the shared (rather than Ras-specific) fundamental mechanisms in glial neoplasia. To support our finding in *Drosophila*, we have performed experiments in multiple human glioma cell lines and observed similar phenotypes that were consistent with those observed in vivo in fly models. It would be important to establish *Drosophila* models using other high-frequency glioma drivers, and we plan to include them in the future.

In the revision, we have included this point in the discussion, on page 13, in paragraph starts with “We adapted a glial neoplasia in *Drosophila* using the *UAS-Ras85D^v12^* and repo-GAL4 driver system that induces overgrowth of glial cells to mimic glial neoplasia formation.”

Another concern is the authors use of glioma nomenclature. *Drosophila* models do not get true "gliomas" as defined by neuropathological criteria for humans and other animals, so the language in the paper describing their glioma model should change to "neoplastic glia," "glial neoplasia" or add the qualifier "model" in their sentences to better distinguish between human and fly biology for this disease. Flies can only model certain aspects of gliomas, such as the genetic basis of neoplastic glial transformation

We thank the reviewer for the suggestion and took the criticism to heart. We followed the suggestion and corrected the nomenclature to ‘glial neoplasia’ throughout the revised manuscript.

Is NMNAT upregulated in tumor specimens from human glioma patients? The authors could query TCGA or other databases (Human Protein Atlas, Ivy Foundation Atlas, Gliovis, etc) to compare the expression levels of NMNAT mRNA and protein between gliomas of different grade and between gliomas with and without mutations in Ras pathway components. These databases are publicly available and this analysis could be done without any benchwork. Or, if they are able to do benchwork, they could compare protein levels in normal human neural stem cells or astrocytes vs glioma cell lines.

We thank the reviewer for this insightful question. We have queried CBioportal and TCGA databases for NMNAT expression in different types of cancers We found high alterations of NMNAT1 and NMNAT2 in multiple cancers. Especially, we found that NMNAT2 is amplified in multiple cancers. (Figure 10_figure supplement 1-2).

To address this comment regarding NMNAT protein expression levels in glioma, we analyzed NMNAT protein levels in a human astroglia cell line (SVG p12), glioma cell lines T98G, and U87MG, and found that NMNAT1 and NMNAT2 were consistently upregulated in glioma cells compared to normal astrocytes (Figure 3_figure supplement 3).

This study may benefit from additional studies on DNA damage such as a comet assay or even looking at whether there are cell cycle changes as a result of modified p53 function.

We thank the reviewer for the suggestion. Using flow cytometry FACS analysis, we have examined the cell cycle (Figure 3_figure supplement 2) and apoptosis (Figure 3 G, H) of glioma cells with or without NMNAT knockdown. Our results suggest that loss of NMNAT did not affect cell cycle, however significantly increased the percentage of apoptotic and pro-apoptotic cells. These results support our model that NMNAT promotes glioma growth through inhibiting apoptosis. This analysis greatly strengthened our conclusion, and we added the apoptosis results to the main figure (Figure 3 G, H), and the cell cycle results to a new supplementary figure (Figure 3_figure supplement 2).

Figure 2E – Manuscript states that Nmnat-PC increased lethality, but the graph is for survival at different life stages. This is confusing, particularly because some of the samples exceed 100% with the way the error bars are presented. This graph should be reformatted to show % lethal broken down by lethal stage.

We have revised the figure as suggested (Figure 2E).

Figure 3 – Colony formation assays are not sufficient to demonstrate growth defects in the glioma cells. Additional independent methods that support these results should be shown. FACS profiling would be appropriate. Also, at least one other glioma cell line that is Ras-dependent should be tested and results from those cells should be shown as well. Why were T98G cells chosen? What is the rationale for using those cells? Were there any other cells tested that yielded negative results? If so, results from those cells may be informative if they are P53 null (many glioma cell lines are P53 null).

We agree with this criticism and have expanded our studies as suggested to include another human glioma cell line (U87MG). The analyses on U87MG cells showed similar results consistent with our conclusion. The results on U87MG cells were included in Figure 5_figure supplement 2 and Figure 9_figure supplement 1.

As stated in the description to the previous comment above, we carried out FACS profiling analysis on cell cycle and apoptosis of glioma cells with or without NMNAT knockdown. Our results suggest that loss of NMNAT did not affect cell cycle (Figure 3_figure supplement 2), however significantly increased the percentage of apoptotic and pro-apoptotic cells (Figure 3 G, H).

Figure 5B – this is an unusual way to quantify cell death in glioma cells. A more representative way would be to quantify the percentage of caspase-positive cells rather than the intensity of caspase staining per cell, which is a much more transient and variable thing. Perhaps the authors already have this data and can change the parameters of their data analysis?

In Figure 5B, we have analyzed caspase-3 immunofluorescence to show activated caspase-3 levels in cells. To further determine the apoptosis, we have performed FACS profiling on cell apoptosis (Figure 3 G, H).

Figure 7A – I would argue only NMNAT2 overexpression is the only condition that increases p53 PARylation. Band intensity could be quantified to better support the author's conclusions.

We recognize the importance of this concern. As described in response to a similar concern raised by reviewer 1, we have carefully examined the NAD^+^-dependent posttranslational modification of p53, including PARylation and (de)acetylation. We employed multiple complementary approaches for quantitatively analyzing each PTM and the new results were included in Figure 8, Figure 9, and Figure 9_figure supplement 1.

Figure S1: There is no criteria set forth to define pre-tumor vs. glioma areas. Are there molecular markers for these regions? Are those markers conserved between human and flies? The presence of excess cells in some regions of the brain is not really sufficient to distinguish between pre-neoplastic and neoplastic cells in the fly brain.

We thank the reviewer for this comment and took this criticism to heart. We have carefully revised the description with regards to nomenclature and definitions to make it clear and consistent. We have defined the criteria for glial neoplasia and the workflow for quantitative analysis of tumor area in vivo in the beginning in Figure 1 and Figure 1_figure supplement 1.

References

Alano, C. C., Tran, A., Tao, R., Ying, W. H., Karliner, J. S., and Swanson, R. A. (2007). Differences among cell types in NAD(+) compartmentalization: A comparison of neurons, astrocytes, and cardiac myocytes. *Journal of Neuroscience Research, 85*(15), 3378-3385. doi:10.1002/jnr.21479

Brazill, J. M., Cruz, B., Zhu, Y., and Zhai, R. G. (2018). Nmnat mitigates sensory dysfunction in a *Drosophila* model of paclitaxel-induced peripheral neuropathy. *Dis Model Mech, 11*(6). doi:10.1242/dmm.032938

Brazill, J. M., Zhu, Y., Li, C., and Zhai, R. G. (2018). Quantitative Cell Biology of Neurodegeneration in *Drosophila* Through Unbiased Analysis of Fluorescently Tagged Proteins Using ImageJ. *JoveJournal of Visualized Experiments*(138). doi:ARTN e58041 10.3791/58041

Camacho-Pereira, J., Tarrago, M. G., Chini, C. C. S., Nin, V., Escande, C., Warner, G. M.,... Chini, E. N. (2016). CD38 Dictates Age-Related NAD Decline and Mitochondrial Dysfunction through an SIRT3-Dependent Mechanism. *Cell Metabolism, 23*(6), 1127-1139. doi:10.1016/j.cmet.2016.05.006

Cheng, H. L., Mostoslavsky, R., Saito, S., Manis, J. P., Gu, Y., Patel, P.,... Chua, K. F. (2003). Developmental defects and p53 hyperacetylation in Sir2 homolog (SIRT1)-deficient mice. *Proc Natl Acad Sci U S A, 100*(19), 10794-10799. doi:10.1073/pnas.1934713100

Guo, Q., Han, N., Shi, L., Yang, L., Zhang, X., Zhou, Y.,... Zhang, M. (2019). NAMPT: A potential prognostic and therapeutic biomarker in patients with glioblastoma. *Oncol Rep, 42*(3), 963-972. doi:10.3892/or.2019.7227

Lucena-Cacace, A., Otero-Albiol, D., Jimenez-Garcia, M. P., Peinado-Serrano, J., and Carnero, A. (2017). NAMPT overexpression induces cancer stemness and defines a novel tumor signature for glioma prognosis. *Oncotarget, 8*(59), 99514-99530. doi:10.18632/oncotarget.20577

Lucena-Cacace, A., Umeda, M., Navas, L. E., and Carnero, A. (2019). NAMPT as a DedifferentiationInducer Gene: NAD(+) as Core Axis for Glioma Cancer Stem-Like Cells Maintenance. *Frontiers in Oncology, 9*. doi:ARTN 292 10.3389/fonc.2019.00292

Rechsteiner, M., Hillyard, D., and Olivera, B. M. (1976). Turnover at nicotinamide adenine dinucleotide in cultures of human cells. *J Cell Physiol, 88*(2), 207-217. doi:10.1002/jcp.1040880210

Ryu, K. W., Nandu, T., Kim, J., Challa, S., DeBerardinis, R. J., and Kraus, W. L. (2018). Metabolic regulation of transcription through compartmentalized NAD(+) biosynthesis. *Science, 360*(6389). doi:ARTN eaan5780 10.1126/science.aan5780

Sampath, D., Zabka, T. S., Misner, D. L., O'Brien, T., and Dragovich, P. S. (2015). Inhibition of nicotinamide phosphoribosyltransferase (NAMPT) as a therapeutic strategy in cancer. *Pharmacology and Therapeutics, 151*, 16-31. doi:10.1016/j.pharmthera.2015.02.004 van Roermund, C. W., Elgersma, Y., Singh, N., Wanders, R. J., and Tabak, H. F. (1995). The membrane of peroxisomes in *Saccharomyces cerevisiae* is impermeable to NAD(H) and acetyl-CoA under in vivo conditions. *Embo Journal, 14*(14), 3480-3486.

Vaziri, H., Dessain, S. K., Eagon, E. N., Imai, S. I., Frye, R. A., Pandita, T. K.,... Weinberg, R. A. (2001). hSIR2(SIRT1) functions as an NAD-dependent p53 deacetylase. *Cell, 107*(2), 149-159. doi:Doi 10.1016/S0092-8674(01)00527-X

Zhai, R. G., Cao, Y., Hiesinger, P. R., Zhou, Y., Mehta, S. Q., Schulze, K. L.,... Bellen, H. J. (2006). *Drosophila* NMNAT maintains neural integrity independent of its NAD synthesis activity. *PLoS Biol, 4*(12), e416. doi:10.1371/journal.pbio.0040416

[Editors’ note: The authors appealed the second decision. What follows is the authors’ response to the second round of review.]

Reviewer #2:Liu et al. investigate the role of the NAD+ synthases NMNAT1 and NMNAT2 in glioma survival to test whether these enzymes are promising targets for therapeutic interventions. The authors employ a glial neoplasia developing *Drosophila* model and the two human glioma cell line models T98G and U87MG. Their data show that depletion of NMNAT1 and NMNAT2 inhibits and overexpression of the enzymes promotes glioma growth. Moreover, the authors show that depletion of NMNAT_1/2_ increased apoptosis while overexpression of the enzymes inhibited apoptosis upon cisplatin treatment. Mechanistically, the authors propose that NMNAT_1/2_ physically interact with p53 and PARP1 to drive PARylation and deacetylation of p53, ultimately leading to reduced p53 activity.I find substantial parts of the underlying data not convincing (see below). I felt that the authors overinterpreted poor or insufficiently controlled data, jumped to conclusions from ambiguous results, and exaggerated several conclusions.Together, I cannot recommend the manuscript for publication.

We appreciate the reviewer’s critical evaluation of our manuscript. We recognized the issues with the description of data and methods as the reviewer pointed out, and a lack of clarity in the writing of results and discussions, especially the sections on human cell models that have resulted in the reviewer’s criticism. Below, we address each of the main concerns that the reviewer stated above, and at the bottom, we include a point-to-point rebuttal for the specific points and questions that the reviewer listed. The revised text and figures are highlighted in blue font in the manuscript.

We are confident that our revision will resolve the ambiguity and confusion, and present a more comprehensive and inclusive model of the role of NMNAT proteins in promoting glioma progression. We thank the reviewer for bringing up their point of view and the opportunity of improving our manuscript in its accuracy and reach in the field of glioma biology.

Notably, during the preparation and submission of our manuscript, Lee Kraus group (UT Southwestern) has also been working on the role of NMNAT in cancers and just published the paper below in Cell on July 19, 2021.

[Ribosome ADP-ribosylation inhibits translation and maintains proteostasis in cancers. Challa S, Khulpateea BR, Nandu T, Camacho CV, Ryu KW, Chen H, Peng Y, Lea JS, Kraus WL. Cell. 2021 Jul 19:S0092-8674(21)00831-X. doi: 10.1016/j.cell.2021.07.005. Online ahead of print. PMID: 34314702]

This paper showed that NMNAT2-mediated cytosolic NAD^+^ synthesis plays an essential role in ovarian cancer by regulating translation and maintaining protein homeostasis through supporting the catalytic activity of the mono(ADP-ribosyl) transferase (MART) PARP-16, which mono(ADP-ribosyl)ates (MARylates) ribosomal proteins. Their finding of the role of NMNAT in protein (ribosome) ADP-ribosylation directly complements our discovery of the role of NMNAT in NAD^+^ dependent PTM in glioma.

In addition, Daisuke Nakada group (Baylor) published the paper showing NMNAT1 mediated NAD+ metabolism enables acute myeloid leukemia (AML) to evade apoptosis.

[Nuclear NAD^+^ homeostasis governed by NMNAT1 prevents apoptosis of acute myeloid leukemia stem cells. Shi X, Jiang Y, Kitano A, Hu T, Murdaugh RL, Li Y, Hoegenauer KA, Chen R, Takahashi K, Nakada D. Sci Adv. 2021 Jul 21;7(30):eabf3895. doi: 10.1126/sciadv.abf3895. Print 2021 Jul. PMID: 34290089.]

Collectively, our manuscript and these two recent papers reveal a common mechanism of NMNAT in promoting cancer growth. Importantly, our manuscript carried out a side-by-side comparative analysis of the role of nuclear vs. cytosolic NMNAT (NMNAT1 and NMNAT2) in promoting the progression of glioma, a fatal cancer, and identified the stronger effect of nuclear NMNAT and its detailed molecular mechanism. These two recent publications further indicate the significance and timeliness of our work.

I am not fully convinced by the growth data presented by the authors. Publicly available data (www.DepMap.org) do not indicate that glioma cell lines are particularly sensitive to NMNAT1 or NMNAT2 depletion. Controls for si-mediated protein reduction in the cell lines are missing throughout and a second siControl could be helpful.

Regarding the growth data and NMNAT in publicly available datasets (such as DepMap.org), we analyzed DepMap.org as suggested and in a similar manner as in Shi et al. 2021 Sci Adv (above paper) (Author response table 1). We found a weak dependence for NMNAT1 and no significant effect for NMNAT2 in glioma cells lines, U87MG or T98G. As a comparison, we also analyzed the gene effect of NMNAT2 in OVCAR3 cells used in Challa et al. 2021 Cell above and as per the CERES score, found no dependence of NMNAT2 in OVCA3 cells, although the published paper clearly shows a functional effect of NMNAT2 in OVCA3 cell growth in ovarian cancer. Analysis of the specific cell lines used in the Shi et al. 2021 Science Advances paper shows their dependency scores are comparable to ours for NMNAT1 (Author response table 1) .

**Author response table 1. sa2table1:** Gene effect analysis using DepMap. org dataset

Manuscript	Cancer	Cell Line	Gene	Gene effect (CERES)	Expression (Log2(TPM+1))
Liu et al. (2021). bioRxiv.(this manuscript)	Glioma	T98G	NMNAT1NMNAT2	-0.427-0.0177	2.872.29
		U87MG	NMNAT1NMNAT2	-0.1590.156	3.15.74
Challa et al. (2021). Cell	Ovarian	OVCAR3	NMNAT1NMNAT2	-0.2190.0989	3.264.18
Shi et al. (2021).Science Advances	Acute myeloid leukaemia	MOLM13	NMNAT1NMNAT2	-0.33-0.00213	2.310.189
		OCIAML2	NMNAT1NMNAT2	0.0588 0.0624	0.8881.52

This supports the notion that findings of negative or weak gene dependency in DepMap datasets do not, in and of themselves, exclude the gene in question’s possible role in cancer growth and progression, and that other parameters and empirical analyses are required to identify the gene’s function in a specific cancer model.

To further examine disease-relevant role of NMNAT in glioma growth, we analyzed the public glioma cancer dataset GEPIA. A strong negative correlation between NMNAT1 expression and survival can be seen in patients with brain lower grade glioma correlation, where patients of higher NMNAT1 expression showed lower survival rate, either the top 50% expression compared to bottom 50% (Figure 11A) or the top 10% expression level compared to the bottom 10% expression (Figure 11B). In the aggressive form of glioma, glioblastoma multiforme (GBM), from which the T98G and U87MG cell lines were derived, the NMNAT1 high expression level (top 10%) showed strong correlation with more aggressive disease and poorer outcome, while NMNAT2 expression level did not correlate with the survival of the aggressive GBM (Figure 11C, D). The brain glioma patient dataset clearly indicates the strong correlation between high NMNAT1 expression with lower survival and poorer clinical outcome. These clinical data are consistent with our findings that NMNAT1 depletion has stronger adverse effects on glioma cell growth vs. NMNAT2 depletion, and higher levels of NMNAT1 promote glioma progression.

Our study provided the mechanism underlying this clinical observation, and identified the effects of NMNAT specifically on promoting glioma growth, without being an oncogene since NMNAT expression alone did not induce glioma initiation. In human glioma cell models, using loss of function (knockdown) and gain of function (overexpression) experiments (MTT assay, growth curve, colony formation), we found NMNAT is required for human glioma cell proliferation without affecting cell cycle control (Figure 3_figure supplement 2). For overexpression experiments, we used DsRed as expression control and for knockdown experiments, we used scrambled siRNA as controls (Figure 3_table supplement 1).

Revision:

1. We have included the GEPIA dataset analysis in a main figure (Figure 10A-D), and revised the discussion to address the implication in glioma tumor initiation vs. progression.

2. We have provided western blot results showing siRNA-mediated protein reduction to complement the PCR result provided for siRNA efficiency (Figure 3_figure supplement 1D and E).

The PARylation data is not convincing. A methods part is missing that explains how PARylation was analyzed. Total PAR or PAR overlay assay? The latter would be crucial to claim p53 PARylation, the former is critical for statements on PARylation in general. The authors claim that NMNAT_1/2_ form a complex with p53 and PARP1 to facilitate PARylation, but it is unclear whether they refer to only p53 PARylation or also PARylation in general.

We used four approaches to detect and analyze PARylation; (i) anti-PAR dot blotting to probe for total PARylation of all proteins, (ii) western blot analysis of total PAR-proteins in cell lysates, (iii) cell imaging with anti-PAR antibody, and (iv) analysis of PARP1 localization and interaction with NMNAT and p53. We found that (1) NMNAT as NAD^+^ synthase increases cellular total PARylation level (Figure 8), and (2) NMNAT expression enhanced the interaction with PARP1 and p53 (Figure 8). Given the trimeric complex formation among NMNAT, PARP1 and p53, and the upregulation of endogenous PARP1 expression in NMNAT overexpressing cells, our findings suggest a co-regulation of NMNAT and PARP1 to facilitate the NAD^+^-dependent PARylation process and p53 is one of the target proteins of PARylation. These data are consistent with the above-mentioned paper (Challa et al. 2021 Cell), and further expand the significance and relevance of the NMNAT mediated protein modifications in cancers.

Revision:

1. Revised methods description of PARylation analysis (page 22).

2. We revised the conclusion to include the broad effects of PARylation and to clarify p53 as one of the targets of PARylation (page 11-12).

3. Added the anti-PAR western blot data of total cell lysates (Figure 8A and B).

The authors draw conclusions on p53 activity from signals that are known to be regulated by various pathways (e.g. apoptosis). Instead, it would be crucial to test for the expression of p53 targets (e.g. MDM2, BBC3, GADD45A) to investigate p53 activity. Yet, given that the p53 status differs between T98G (mutant p53) and U87MG (wild-type p53) cells (van Meir et al., 1994 Cancer Res), the authors' growth data, which are similar for T98G and U87MG, demonstrate that the potential effect elicited by NMNAT_1/2_ is independent of. Moreover, all p53 interaction data shown using T98G data reflect mutant p53, which is known to have potentially different binding partners than wild-type p53. In general, the fact that T98G harbor mutant p53 turns most parts of the manuscript upside down that deal with mechanistics.

Regarding p53 activity and the p53 status in T98G and U87MG cells. Our findings of similar effects of NMNAT on p53 modification in either wild-type (U87MG) or mutant p53 (T98G) cells, suggest NAD^+^-dependent PARylation or deacetylation of p53 is independent of the p53 [M237I] mutation. This is a gain-of-function mutation in the DNA binding domain that retains the ability of p53 to be modified by acetylation and PARylation as the mutation does not affect the sites of these PTMs (Yamamoto and Iwakuma, 2018; Yi et al., 2013). Furthermore, there is ample literature characterizing apoptosis induction in the T98G line, including through p53-dependent mechanisms (eg. (Enns, Bogen, Wizniak, Murtha, and Weinfeld, 2004) shows the p53 inhibitor pifithrin can inhibit apoptosis in irradiated T98G cells). Indeed, recent studies have shown that mutant p53 can retain the ability to induce apoptosis despite losing tumor-suppressive transactivation functionality (Timofeev et al., 2019). The Timofeev et al. study is one of several lines of emerging data on the complexity of p53 mutants with regards to their retained ability to enact similar functions as wildtype p53, and it can no longer be generalized that mutant p53 will necessarily have different binding partners than wild-type p53, in the face of contrary evidence (as provided by us here with regards to NMNAT p53 interactions).

Because approximately 51% of glioma are mutated for p53, we intentionally designed a comparative analysis using two different glioma cell lines with different p53 status to dissect common mechanisms of the role of NMNAT in glioma cell growth through modulating global NAD^+^-dependent protein modification. The common apoptosis-inhibitory effect of NMNAT expression we report on both cell lines does not exclude NAD^+^-dependent modifications on proteins (other than p53) that play important tumor growth-regulatory roles, in addition to p53. The ribosomal proteins shown in the above-mentioned paper (Challa et al. 2021 Cell) are a potential example. Regardless, our use of lines with differing p53 mutational status here is a strength, showing the pervasive utility of targeting NMNAT, rather than a weakness of our study.

What our findings collectively show for the first time is a role for NMNAT in glioma progression through regulating the NAD^+^-dependent post-translational modifications of proteins PARylation and deacetylation, with p53 as one of the targets.

Figure 1: why is Nmnat not color-coded?

We used gray scale for Nmnat channel in Figure 1 in order to highlight the upregulation of NMNAT protein in *Ras^v12^* overexpressing glial cells.

Figure 3G: The AnnexinV:PI gating is strange. For comparison see https://images.bio-rad-antibodies.com/kit/annexin-v-kit-antibody-kit-annex100-image2-600w.jpg.

We used the apoptosis detection kit from BD Biosciences and followed the specific instructions (https://www.bdbiosciences.com/en-us/products/reagents/flow-cytometry-reagents/research-reagents/panels-multicolor-cocktails-ruo/fitc-annexin-v-apoptosis-detection-kit-i.556547). The AnnexinV:PI gating was selected to divide the data in quadrants, where Q3 was considered as viable, Q4 as early apoptosis and Q2 as end stage apoptosis and death. The Figure 3H indicates the sum of Q4 and Q2 of each group. We originally segmented the P4 and P5 boxes in Q2 to highlight the apoptotic cell population that increased dramatically after siRNA mediated NMNAT reduction. However, since we included the entire Q2 in quantification, we have removed P4 and P5 segments to improve the clarity of the results.

Revision: We have removed P4 and P5 segmentation (Figure 3G) and provided the details of analysis including gating criteria in the methods section and figure legends (page 8 and page 23).

Western blot signals are often saturated, which hinders proper readout.Figures 5A, B, D, and F miss NMNAT1 and NMNAT2 as crucial controls.

We have provided the mRNA level efficiency of siRNA (Figure 3_figure supplement 1C).

Revision: We have added the western analysis of NMNAT protein level (Figure 3_figure supplemental 1D and E).

Figures 5D and F: All lanes display cisplatin-treated samples? If so, controls without cisplatin treatment are missing.

In this Figure, cisplatin was used to induce apoptosis only in the NMNAT overexpression experiment (Figure 5B) to observe the potential reduction of apoptosis by NMNAT. In NMNAT knockdown experiment (Figure 5A and 5D), cisplatin is not used because we expect to observe a potential increase with NMNAT knockdown.

Revision: To clarify this point, we have included the detailed information in Figure 5 figure legend and main text (page 9).

Figure 8A and Figure 8—figure supplement 1: PARylation of what? Controls are missing.

Figure 8A detected PARylation of total protein in cell extracts using anti-PAR antibody. We determined protein concentration and loaded same amount proteins of each group. DsRed expressing cells were used as controls.

Revision: We have included the detailed information in Figure 8 figure legend and main text (page 11).

Figure 8C: Specificity of the NMNAT2 antibody is not convincing.

We used the NMNAT2 specific antibody (Abcam AB56980). We identified the NMNAT2 protein bands by the NMNAT2 overexpression lane (Figure 8C input lane).

Figure 9A: It is unclear why the authors IP p53 to blot acetyl-p53. Acetyl-p53 can be immunoblotted with whole cell lysates, as demonstrated by the authors (input lanes). Loading of IPs is more difficult to control, thus the signal from the input lanes are more important and actually do not show changes in acetyl-p53 upon NMNAT overexpression.

We used p53 IP experiment to serve two purposes; (i) to probe the potential interaction of p53 with deacetylation enzyme SIRT1, and (ii) to clearly identify the acetylated-p53 protein bands. Indeed as the reviewer suggested, we can quantify using the input lanes in addition to the IP-ed p53.

Revision: We have quantified acetyl-p53 in the input lanes (Figure 9C).

Figure 9C: What is quantified here? 9B apparently is based on the p53 IP lanes. But 9C appears to be based on the input lanes. Cherry picking?

Figure 9B quantifies the acetyl-p53 immunoprecipitated by anti-p53 antibody. Figure 9C quantifies the endogenous level of SIRT1 in cell lysates as in the input lanes. Since we did not detect any interaction between p53 and SIRT1, we determined the level of endogenous SIRTs in input. As indicated in the response to last point, we IP-ed p53 to avoid the nonspecific bands shown in the total lysate. The quantifications included in this figure were justified given the experimental feasibility. We strongly believe the criticism of ‘cherry picking’ is unfounded.

Revision: We have quantified acetyl-p53 in the input lanes (Figure 9C).

Figure 3—figure supplement 3: NMNAT2 blot is out of focus.

Our western analyses were all carried out using Li-COR infrared fluorescence scans. There should be no ‘focusing’ involved.

Revision: we have replaced the figure with a full blot with better appearance (Figure 3_figure supplement 3).

Figure 5—figure supplement 2: reduction of cleaved cas3 upon NMNAT1 overexpression is not convincing. siNMNAT_1/2_ are missing. Immunoblots for NMNAT_1/2_ are missing.

Figure 5 and the accompanying supplemental figures were addressing the effect of overexpression NMNAT in caspase-mediated apoptosis. We have shown in the previous figures that loss of NMNAT_1/2_ result in cell death.

Figure 9—figure supplement 1: What is shown in this Figure? Whole cell lysates or p53 IPs? Why do the authors use only lysate or IP here but both in Figure 9A? The reduction is not convincing. No cisplatin control is missing. siNMNAT_1/2_ are missing.Immunoblots for NMNAT_1/2_ are missing.

In Figure 9—figure supplement 1, we used whole cell lysates as indicated in the figure legend. Figure 9A is on T98G cells where p53 (mutant) is expressed at a high level. However, the U87MG cells have low expression level of p53 (wildtype) as shown in Figure 5_supplemnet 2. Cisplatin treatment was used only used in U87MG cells to induce p53 expression level to allow the analysis of the posttranslational modification of p53.

References

Enns, L., Bogen, K. T., Wizniak, J., Murtha, A. D., and Weinfeld, M. (2004). Low-dose radiation hypersensitivity is associated with p53-dependent apoptosis. *Molecular Cancer Research, 2*(10), 557-566. Retrieved from <Go to ISI>://WOS:000224650800004

Timofeev, O., Klimovich, B., Schneikert, J., Wanzel, M., Pavlakis, E., Noll, J.,… Stiewe, T. (2019). Residual apoptotic activity of a tumorigenic p53 mutant improves cancer therapy responses. *Embo Journal, 38*(20), e102096. doi:10.15252/embj.2019102096

Yamamoto, S., and Iwakuma, T. (2018). Regulators of Oncogenic Mutant TP53 Gain of Function. *Cancers (Basel), 11*(1). doi:10.3390/cancers11010004

Yi, Y. W., Kang, H. J., Kim, H. J., Kong, Y., Brown, M. L., and Bae, I. (2013). Targeting mutant p53 by a SIRT1 activator YK-3-237 inhibits the proliferation of triple-negative breast cancer cells. *Oncotarget, 4*(7), 984-994. doi:10.18632/oncotarget.1070

[Editors’ note: what follows is the authors’ response to the third round of review.]

There is consensus that the manuscript has been much improved but there are some remaining issues that need to be addressed, as outlined below:We would like you to submit a final paper that includes:1) Data indicating that NMNAT_1/2_ interacts with p53 also in the U87MG p53 wild-type line;

We have carried out the immunoprecipitation experiment as suggested and detected the interaction between p53 and NMNAT_1/2_ in U87MG cells. Using an anti-p53 antibody, we immunoprecipitated p53 from U87MG cell lysates and probed for NMNAT1 and NMNAT2. As shown in the figure included, we detected NMNAT1 and 2 in the p53-IP fraction, suggesting that NMNAT_1/2_ interacts with wildtype p53. We have added this figure as Figure 9_figure supplement 2.

2) Data that tests if p53 depletion rescues the apoptosis-inducing effect of siNMNAT in both cell lines they use (T98G and U87MG);

To address this point, we have used two approaches to reduce/deplete p53: siRNA transfection or shRNA lentivirus transduction in both cell lines. After p53 depletion, we carried out siNMNAT knockdown and probed for cleaved caspase3 to examine the activation of apoptosis in both T98G and U87MG cells. Under all four conditions, two cell types and two modes of p53 depletion, we observed consistent reduction of siNMNAT-induced apoptosis activation when p53 is depleted. As shown in the figures included, cleaved caspase3 expression level in p53 and NMNAT double knockdown cells was reduced compared to those in siNMNAT cells. These observations support that p53 is required for the apoptosis-inducing effect of siNMNAT in both cell lines. We have included the data with shRNA lentivirus transduction as a new figure (Figure 8) and the data with siRNA transfection as a supplemental figure (Figure 8—figure supplement 1).

3) Toning down claims on PARylation at different locations in the manuscript.

We recognize the importance of accurate interpretation of our results and, following this suggestion, we have revised the entire manuscript. Specifically, we have made the following changes.

1) Abstract. We have revised the sentence, “Interestingly, NMNAT forms a complex with p53 and PTM enzyme PARP1 to facilitate PARylation. PARylation and deacetylation reduce p53 pro-apoptotic activity, indicating that regulation of p53 post-translational modifications is a key mechanism by which NMNAT promotes glioma growth” to “Since PARylation and deacetylation reduce p53 pro-apoptotic activity, modulating p53 post-translational modifications could be a key mechanism by which NMNAT promotes glioma growth”.

2) Figure 9. We revised the title “NMNAT interacts with PARP1 and upregulates PARylation of p53” to “NMNAT interacts with p53 and PARP1 and upregulates PARylation”.

3) Results. We revise the conclusion in the paragraph about PARylation, line 289, “Collectively, these results suggest NMNAT regulates p53 modification by complexing with p53 and PARP1, thus potentially increasing the local NAD^+^ availability to promote PARylation with high efficiency” to “Collectively, these results suggest NMNAT interacts with PARP1 and promotes PARylation of PARP1 targeting proteins, like p53, potentially through increasing the local NAD^+^ availability”.

4) Discussion. We revised the sentences in the first paragraph, line 329, “Mechanistically, upregulation of enzymatically active NMNAT promotes the NAD^+^-dependent post-translational modifications of p53, and specifically increases the PARylation of p53 and reduces the acetylation of p53. Furthermore, we detected presence of a p53-NMNAT-PARP1 trimeric complex as well as increased SIRT1, suggesting a highly efficient NAD^+^-dependent post-translational modification process facilitated by the NAD^+^ synthase function of NMNAT” to “Mechanistically, upregulation of enzymatically active NMNAT promotes the NAD^+^-dependent post-translational modifications of p53. Specifically, we detected upregulation of protein PARylation and the presence of a p53-NMNAT-PARP1 trimeric complex and decreased acetylation of p53 accompanied with increased SIRT1”.

Reviewer #2:The authors provided a revised version addressing most concerns that I raised. However, some key points in the authors' study still require supportive data.– The NMNAT-PARP1-p53 interaction:Previously, I raised the concern that protein interaction with mutant p53 not always translates into an interaction with wild-type p53. The present M237I mutant reportedly possesses neomorphic functions. Given that the NMNAT_1/2_-p53 interaction is an integral part of the model proposed by the authors, I would like to reiterate that I find it crucial to corroborate this interaction also in the p53 wild-type line U87MG.

We have carried out the immunoprecipitation experiment as suggested and detected the interaction between p53 and NMNAT_1/2_ in U87MG cells. Using anti-p53 antibody, we immunoprecipitated p53 from U87MG cell lysates and probed for NMNAT1 and NMNAT2. We detected NMNAT1 and 2 in the p53-IP fraction, suggesting the interaction of NMNAT_1/2_ with wildtype p53.

Revision: we have added a new figure to show this result (Figure 9_figure supplement 2).

– The NMNAT-p53-apoptosis mechanism:Following my concern that much of the authors' data on p53 was generated using a mutant p53 cell line (T98G), the authors explain that they intentionally included both wild-type and mutant p53 cell systems and clarify their strategy to the reader in the revised version, such as by adding a respective sentence to the Discussion section "Our findings that NMNAT similarly affects p53 modification in either wild-type (U87MG) or mutant p53 (T98G) cells suggest NAD+-dependent PARylation or deacetylation of p53 is independent of the p53 [M237I] mutation. Indeed recent studies have shown that mutant p53 proteins retain the ability to induce apoptosis despite losing tumor-suppressive transactivation functionality (Timofeev et al., 2019)".Notably, Timofeev et al., 2019 refers to the p53 mutant R181E (R178E in mice). p53 mutants are known to differ. The M237I mutant present in the authors' T98G cell line, for example, did not display residual apoptosis-driving function when it was tested in a different study (Boettcher et al., 2019 Science). Given that the authors propose reduced p53-dependent apoptosis to be a key mechanism by which NMNAT promotes glioma growth, it is crucial to show that there actually is p53-dependent apoptosis occurring in the authors' experimental setup, i.e. by adding data on sip53 in Figure 3G showing whether p53 depletion can indeed rescue the apoptosis-inducing effect of siNMNAT in both T98G and U87MG cells. Given the importance to the authors' mechanistic model and the different experimental setup, it is insufficient to only refer to the findings by Enns et al., 2004.

We appreciate the reviewer’s concern. To address this point, we have used two approaches to reduce/deplete p53: siRNA transfection or shRNA lentivirus transduction in both cell lines. After p53 depletion, we carried out siNMNAT knockdown and probed for cleaved caspase3 to examine the activation of apoptosis in both T98G and U87MG cells. Under all four conditions, two cell types and two modes of p53 depletion, we observed consistent reduction of siNMNAT-induced apoptosis activation by p53 depletion. As shown in the figures included, cleaved caspase3 expression level in p53 and NMNAT double knockdown cells was reduced compared to that in siNMNAT cells. These observations support that p53 is required for the apoptosis-inducing effect of siNMNAT in both cell lines. We have included the data with shRNA lentivirus transduction as a new figure (Figure 8) and the data with siRNA transfection as a supplemental figure (Figure 8—figure supplement 1).

Revision: we have added two new figures to show the result (Figure 8 and Figure 8_figure supplement 1).

– The NMNAT-p53-PARylation mechanism:The authors convincingly demonstrate that NMNAT_1/2_ affect total PAR levels in the cell (Figure 8A and B, Figure 8-supplement 1). Key points in the authors' model include (1) that PARylation of p53 is induced by NMNAT_1/2_ (abstract, headlines, Figure 10E) and (2) that complex formation with p53 is important for NMNAT_1/2_ to facilitate PARylation (abstract). Supportive data for these points, however, is missing.1) I would like to reiterate that it is crucial to provide data on NMNAT-dependent p53 PARylation. I.e. by blotting for PAR in p53 IPs (Figure 8C or 9A), in both T98G and U87MG.

We thank the reviewer for the suggestion. Due to their transient nature, the PARylation of p53 could be difficult to detect consistently. Since there is no antibody available to specifically detect PAR-p53, we used two approaches to address this question. First, we assessed total PARylation level in cells with or without NMNAT overexpression and found that total PARylation is increased with NMNAT overexpression (Figure 9 A, B), suggesting that NMNAT promotes protein PARylation in general. Second, to probe the probability of p53 PARylation, we immunoprecipitated p53 and probed for the PARylation enzyme PARP1 (Figure 9C). We found that the p53, PARP1, and NMNAT form a trimeric complex that was stable in immunoprecipitation both in T98G and U87MG. Importantly, we found that endogenous PARP1 expression was upregulated in NMNAT overexpressing cells suggesting a co-regulation of NMNAT and PARP1 to facilitate the NAD^+^-dependent PARylation. These biochemical results were corroborated by the immunofluorescent imaging analysis where cells overexpressing NMNAT showed a higher level of PARP1 colocalizing with p53 (Figure 9 D-I). Together, these results support our hypothesis that p53 PARylation is increased in NMNAT overexpressing cells, although we acknowledge that future studies with specific antibodies to directly detect the p53 PARylation is required for confirmation.

2) To support the authors' point that "NMNAT forms a complex with p53 and PTM enzyme PARP1 to facilitate PARylation" (abstract), it is crucial to show whether p53 indeed is required for NMNAT-dependent PARylation, i.e. whether p53 depletion affects NMNAT-dependent PARylation in both T98G and U87MG (Figures 8A and 8-supplement 1).

Our findings suggest NMNAT interacts with PARP1 to facilitate NAD^+^-dependent PARP1-mediated PARylation, and p53 is likely one of many target proteins of NMNAT-dependent PARylation. Whether p53 regulates NMNAT-dependent PARylation would be an interesting topic for future investigation.

We recognize the importance of accurate interpretation of our results and followed this suggestion and have revised the entire manuscript. We have toned down our conclusions as follows:

1) we revise the sentence in abstract, “Interestingly, NMNAT forms a complex with p53 and PTM enzyme PARP1 to facilitate PARylation. PARylation and deacetylation reduce p53 pro-apoptotic activity, indicating that regulation of p53 post-translational modifications is a key mechanism by which NMNAT promotes glioma growth” to “Since PARylation and deacetylation reduce p53 pro-apoptotic activity, modulating p53 post-translational modifications could be a key mechanism by which NMNAT promotes glioma growth”.

2) we revise the Figure 9 title “NMNAT interacts with PARP1 and upregulates PARylation of p53” to “NMNAT interacts with p53 and PARP1 and upregulates PARylation”.

3) we revise the conclusion in the paragraph of Results about PARylation, page 12, “Collectively, these results suggest NMNAT regulates p53 modification by complexing with p53 and PARP1, thus potentially increasing the local NAD^+^ availability to promote PARylation with high efficiency” to “Collectively, these results suggest NMNAT interacts with PARP1 and promotes PARylation of PARP1 targeting proteins, like p53, potentially through increasing the local NAD+ availability”.

4) we revise the sentences in the first paragraph of Discussion, page 14, “Mechanistically, upregulation of enzymatically active NMNAT promotes the NAD^+^-dependent post-translational modifications of p53, and specifically increases the PARylation of p53 and reduces the acetylation of p53. Furthermore, we detected presence of a p53-NMNAT-PARP1 trimeric complex as well as increased SIRT1, suggesting a highly efficient NAD^+^-dependent post-translational modification process facilitated by the NAD^+^ synthase function of NMNAT” to “Mechanistically, upregulation of enzymatically active NMNAT promotes the NAD^+^-dependent post-translational modifications of p53. Specifically, we detected upregulation of protein PARylation and the presence of a p53-NMNAT-PARP1 trimeric complex and decreased acetylation of p53 accompanied with increased SIRT1”.